EMBO
Molecular Medicine

# Asymptomatic COVID-19: disease tolerance with efficient anti-viral immunity against SARS-CoV-2

Yi-Hao Chan[1,2,†] [iD], Siew-Wai Fong[1,2,†] [iD], Chek-Meng Poh[1,2,†] [iD], Guillaume Carissimo[1,2,†] [iD], Nicholas Kim-Wah Yeo[1,2] [iD], Siti Naqiah Amrun[1,2] [iD], Yun Shan Goh[1,2], Jackwee Lim[2] [iD], Weili Xu[2] [iD], Rhonda Sin-Ling Chee[1,2] [iD], Anthony Torres-Ruesta[1,2,3] [iD], Cheryl Yi-Pin Lee[1,2] [iD], Matthew Zirui Tay[1,2] [iD], Zi Wei Chang[1,2] [iD], Wen-Hsin Lee[2] [iD], Bei Wang[2] [iD], Seow-Yen Tan[4] [iD], Shirin Kalimuddin[5,6] [iD], Barnaby Edward Young[7,8,9] [iD], Yee-Sin Leo[7,8,9,10,11] [iD], Cheng-I Wang[2] [iD], Bernett Lee[2] [iD], Olaf Rötzschke[2] [iD], David Chien Lye[7,8,9,11] [iD], Laurent Renia[1,2,*] [iD] & Lisa F P Ng[1,2,3,12,*] [iD]

## Abstract

The immune responses and mechanisms limiting symptom progression in asymptomatic cases of SARS-CoV-2 infection remain unclear. We comprehensively characterized transcriptomic profiles, cytokine responses, neutralization capacity of antibodies, and cellular immune phenotypes of asymptomatic patients with acute SARS-CoV-2 infection to identify potential protective mechanisms. Compared to symptomatic patients, asymptomatic patients had higher counts of mature neutrophils and lower proportion of CD169[+] expressing monocytes in the peripheral blood. Systemic levels of pro-inflammatory cytokines were also lower in asymptomatic patients, accompanied by milder pro-inflammatory gene signatures. Mechanistically, a more robust systemic Th2 cell signature with a higher level of virus-specific Th17 cells and a weaker yet sufficient neutralizing antibody profile against SARS-CoV-2 was observed in asymptomatic patients. In addition, asymptomatic COVID-19 patients had higher systemic levels of growth factors that are associated with cellular repair. Together, the data suggest that asymptomatic patients mount less pro-inflammatory and more protective immune responses against SARS-CoV-2 indicative of disease tolerance. Insights from this study highlight key immune pathways that could serve as therapeutic targets to prevent disease progression in COVID-19.

Keywords asymptomatic; COVID-19; disease tolerance; SARS-CoV-2
Subject Categories Immunology; Microbiology, Virology & Host Pathogen Interaction

## Introduction

Coronavirus disease 2019 (COVID-19) is a respiratory disease caused by severe acute respiratory syndrome coronavirus 2 (SARS-CoV-2) that was first detected in Wuhan, China, in late 2019 (Zhu et al, 2020). Infection causes a wide variety of clinical manifestations, ranging from asymptomatic infection to acute respiratory distress syndrome (ARDS) and death (Baud et al, 2020; Chaudhry et al, 2020; Guan et al, 2020; Yang et al, 2020; Zhou et al, 2020). During viral infections, the host immune response is the key determinant in virus clearance and resolution of disease. Several risk factors, including age, existing co-morbidities, or ongoing immunosuppressive therapy, disrupt a well-regulated anti-viral response and lead to detrimental outcomes (Harrison et al, 2020; Zhou et al, 2020). In severe infections, this includes lymphopenia (Tan et al, 2020), dysregulated myeloid compartment (Carissimo et al, 2020;

1   A*STAR Infectious Diseases Labs (A*STAR ID Labs), Agency for Science, Technology and Research, Singapore City, Singapore
2   Singapore Immunology Network, Agency for Science, Technology and Research, Singapore City, Singapore
3   Department of Biochemistry, Yong Loo Lin School of Medicine, National University of Singapore, Singapore City, Singapore
4   Department of Infectious Diseases, Changi General Hospital, Singapore City, Singapore
5   Department of Infectious Diseases, Singapore General Hospital, Singapore City, Singapore
6   Emerging Infectious Diseases Program, Duke-NUS Medical School, Singapore City, Singapore
7   National Centre for Infectious Diseases, Singapore City, Singapore
8   Department of Infectious Diseases, Tan Tock Seng Hospital, Singapore City, Singapore
9   Lee Kong Chian School of Medicine, Nanyang Technological University, Singapore City, Singapore
10  Saw Swee Hock School of Public Health, National University of Singapore and National University Health System, Singapore City, Singapore
11  Yong Loo Lin School of Medicine, National University of Singapore, Singapore City, Singapore
12  Institute of Infection, Veterinary and Ecological Sciences, University of Liverpool, Liverpool, Liverpool, UK
    *Corresponding author. Tel: +65 6407 0005; E-mail: renia_laurent@IDLabs.a-star.edu.sg
    **Corresponding author. Tel: +65 6407 0028; E-mail: Lisa_Ng@IDLabs.a-star.edu.sg
    †These authors contributed equally to this work

Silvin *et al*, 2020), and hyper-inflammatory responses (Manson *et al*, 2020; Young *et al*, 2020; Zhou *et al*, 2020).

The high proportion of asymptomatic SARS-CoV-2 infections (estimated at 30–40%) (Ng *et al*, 2020) has led to unprecedented challenges for containing the pandemic. These infections can be significant, but undetected, causes of community transmission without routine screening. These cases are also scientifically intriguing as they demonstrate disease tolerance, illustrated with an immune response that efficiently resolves the viral infection without apparent pathogenic effects. Thus, understanding the immune responses of asymptomatic patients may be relevant for developing immune-modulatory treatments to ameliorate symptom progression in COVID-19 patients and may also improve our understanding of whether COVID-19 vaccines offer protection against infection and transmission.

It has been proposed that asymptomatic patients mount a potent innate immune response protective against SARS-CoV-2 infection (preprint: Zhao *et al*, 2020). However, most of these studies have investigated the memory-like immune response in convalescent samples, which may not reflect the immune state during acute SARS-CoV-2 infection. Counterintuitively, observations in patient convalescent plasma showed positive correlations between antibody levels and disease severity (Amrun *et al*, 2020), suggesting that asymptomatic patients trigger a milder adaptive immune response (Long *et al*, 2020). In this study, we present a comprehensive study of the transcriptomic profiles, cytokine responses, neutralization profiles of patients' sera, and high-density immunophenotyping during active infection between asymptomatic and symptomatic patients in the Singapore COVID-19 cohort. Our data show that asymptomatic patients mount a balanced adaptive cellular immune response that is less pro-inflammatory and yet protective, indicative of acquisition of disease tolerance to the infection.

## Results

### Asymptomatic COVID-19 patients demonstrate distinct transcriptomic profiles from symptomatic patients during active infection

Asymptomatic COVID-19 patients from this cohort study were mostly young Bangladeshi, Filipino, and Myanmar workers from the dormitory outbreaks compared to majority of Chinese among symptomatic patients, and a similar proportion of patients in both groups had associated co-morbidities (Table EV1). In addition, the initial cycle threshold (Ct) values for polymerase chain reaction assay in asymptomatic patients were higher than symptomatic patients, indicating lower viral loads in nasopharyngeal swabs at the point of diagnosis (Table EV1). Notably, symptomatic patients had significantly higher C-reactive protein and lactate dehydrogenase levels compared to asymptomatic patients, indicating higher levels of inflammation (Poggiali *et al*, 2020) (Table EV1).

We studied 48 asymptomatic COVID-19 patients and compared their transcriptomic signatures, soluble immune mediator levels, and immune cell profiles during acute infection against 172 symptomatic patients (Table EV2). High-throughput RNA-sequencing was performed to comprehensively characterize transcriptomic profiles of asymptomatic (median 3 days post-admission, interquartile range (IQR) 2-3) and symptomatic (median 8 days post-illness onset (PIO),

IQR 4-11) COVID-19 patients during acute infection. Twenty-six symptomatic patients with varying disease severity outcomes (mild, symptomatic without pneumonia, $n = 9$; moderate, pneumonia without oxygen requirement, $n = 10$; and severe, pneumonia with oxygen requirement, $n = 7$) and 30 asymptomatic patients with RNA integrity number more than 6 were included in this analysis (Table EV2). Due to the differences in timing of diagnosis and hospitalization, blood sample collections were performed at varying stages of the acute phase, which could influence the observed transcriptomic profiles in symptomatic patients. Nevertheless, we observed 215 differentially expressed genes (DEGs) under-expressed and 952 over-expressed in symptomatic patients (Fig 1A, Dataset EV1) during acute phase of infection when we compared symptomatic and asymptomatic patients at a threshold of false discovery rate (FDR)-adjusted $P < 0.05$ and fold change (FC) > 2. Principal component analysis (PCA) also showed a clear distinction between the two groups of patients (Fig 1B). Gene Ontology (GO) enrichment, which allows the elucidation of the biological processes of the involved DEGs, revealed that asymptomatic COVID-19 patients had less robust response to type-I interferon, classical complement, innate, and humoral immune responses, but presented with upregulation of processes such as cytosolic ribosomal activity, positive regulation of cell killing, T-cell, and TNF receptor activities, and regulation of B-cell proliferation during the acute phase of SARS-CoV-2 infection (Fig 1C). Overall, the transcriptomic data suggest that differences between asymptomatic and symptomatic patients during acute SARS-CoV-2 infection may be present at the cellular, innate, and adaptive immune response levels.

### Dampened systemic inflammation and higher levels of growth factors in asymptomatic patients

To validate the transcriptomic responses observed, we assessed the immune soluble mediators in acute plasma samples from asymptomatic ($n = 48$) and symptomatic ($n = 172$) patients at the first collection timepoint upon hospital admission with a 45-plex microbead-based immunoassay (Thermo Fisher Scientific) (Table EV2). These were further stratified into their severity stratum—mild patients ($n = 61$), moderate patients ($n = 43$), and severe patients ($n = 68$) (Table EV2). Collectively, we observed higher concentrations of growth factors BDNF and VEGF-D in asymptomatic patients (median 3 days post-admission, IQR 2–3) compared to symptomatic patients (median 8 days post-illness onset, IQR 4-12.75) (Figs 2A and EV1A). In addition, asymptomatic patients had lower levels of pro-inflammatory cytokines (IFN-γ, IL-1β, IL-18, IL-6, IL-7) and chemokines (IP-10 and MCP-1), along with lower levels of markers associated with lung injury and regeneration (HGF, VEGF-A, and EGF) (Figs 2A and EV1B and C). Protein–protein interaction network analyses with STRING (Search Tool for the Retrieval of Interacting Genes/Proteins) highlighted multiple direct and indirect interactions between the significant immune mediators and their involvement in biological processes such as T cell-mediated immunity (IL-2, IL-6, IL-12p70, IL-18, IFN-γ), inflammation (IL-5, IL-6, IL-18, IP-10, MIP-1β, IL-1RA, IL-17A), chemotaxis (IL-6, MIP-1β, IP-10, BDNF, HGF, VEGF-A, VEGF-D, PDGF-BB), and angiogenesis (IP-10, VEGF-A, VEGF-D, HGF, PlGF) (Fig 2B). Together, these results suggest that the T-cell immunity and pro-inflammatory responses are differentially regulated in asymptomatic patients, suggesting reminiscence of disease tolerance.

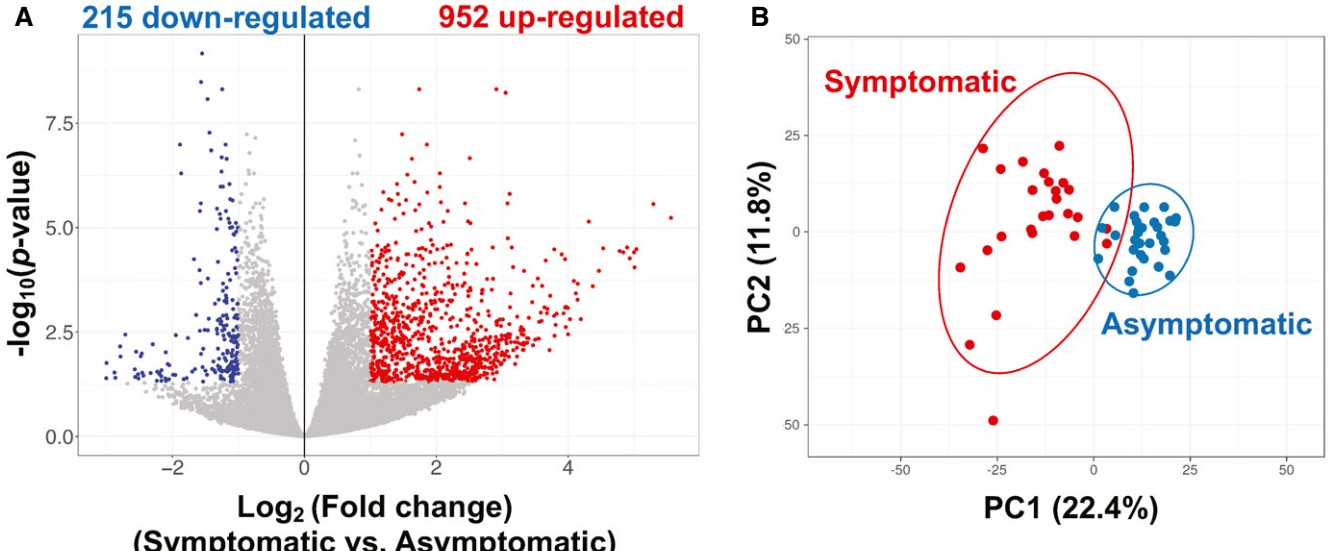

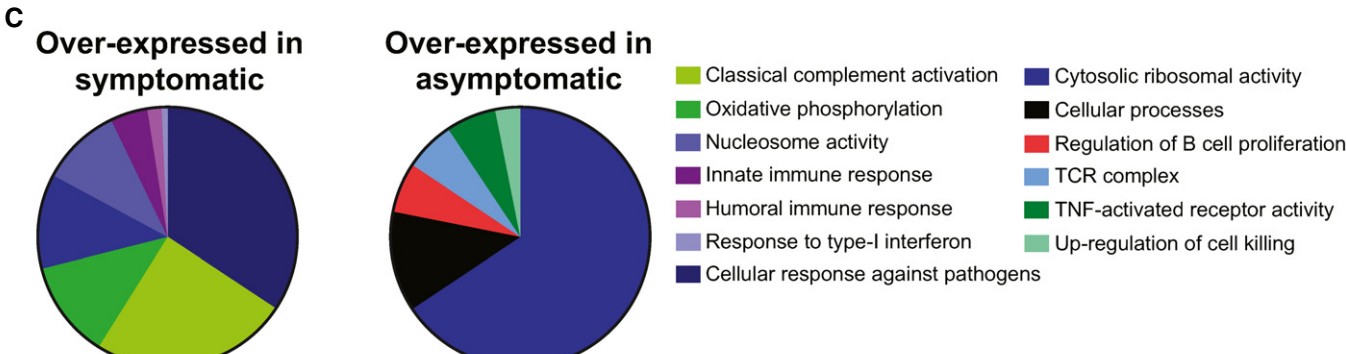

**Figure 1. Whole blood transcriptome analysis in symptomatic and asymptomatic COVID-19 patients highlights distinctive immune signatures during the acute phase of infection.**

A   RNA-seq of whole blood from symptomatic ($n = 26$; median 8 days PIO) and asymptomatic ($n = 30$; median 3 days post-hospital admission) COVID-19 patients at the acute phase of infection (SARS-CoV-2 PCR-positive) was performed. Only samples with RNA integrity number > 6 were sent for sequencing and included in the analysis. Volcano plot indicating DEGs between asymptomatic and symptomatic COVID-19 patients, with thresholds of FDR < 0.05 and |FC| > 2. Numbers of over-expressed and under-expressed genes are indicated.

B   PCA of symptomatic and asymptomatic COVID-19 patients based on DEGs, with FDR < 0.05 and |FC| > 2.

C   Pie chart showing distribution of the functionally grouped GO terms using ClueGO. Illustrated genes were under-expressed or over-expressed in asymptomatic patients compared to symptomatic patients. Each GO term is statistically significant (Benjamini–Hochberg correction < 0.05), with a cutoff set at $P < 0.05$.

## Asymptomatic patients exhibit robust virus-specific Th17 responses

Systemic levels of T cell-associated cytokines were further analyzed in severity-stratified patients to investigate the possibility of a Th1/Th2 imbalance influencing COVID-19 severity (Fig EV1). Interestingly, Th1-associated cytokines IFN-γ, IL-1β, and IL-18 were significantly lower in asymptomatic COVID-19 patients, while IFN-γ and IL-18 increased with disease severity (Fig EV1). In addition, asymptomatic patients had lower systemic levels of Th2-associated IL-5, although there were no significant differences between symptomatic patients of different severity stratum (Fig EV1).

Corroborating the differences in soluble mediator levels, transcriptomic profiles of COVID-19 patients also revealed a clear distinction in T cell-associated signatures between symptomatic and asymptomatic patients (Fig 3A). These DEGs were categorized and presented as specific T effector cell-associated signatures based on previously reported T-cell signatures in blood transcriptomes of COVID-19 patients (Aschenbrenner et al, 2021) (Fig 3A). GO enrichment and ingenuity pathway analysis were also performed to functionally categorize these DEGs (Fig 3A). DEGs showed an upregulation of genes involved in Th2 pathway, T-cell proliferation, PD-1 signaling, and TCR signaling, together with a lower expression of genes associated with Th1 and Th17 pathways in asymptomatic patients.

To validate these observations, phenotyping of naïve and differentiated cells during acute infection was performed and compared between symptomatic and asymptomatic patients.

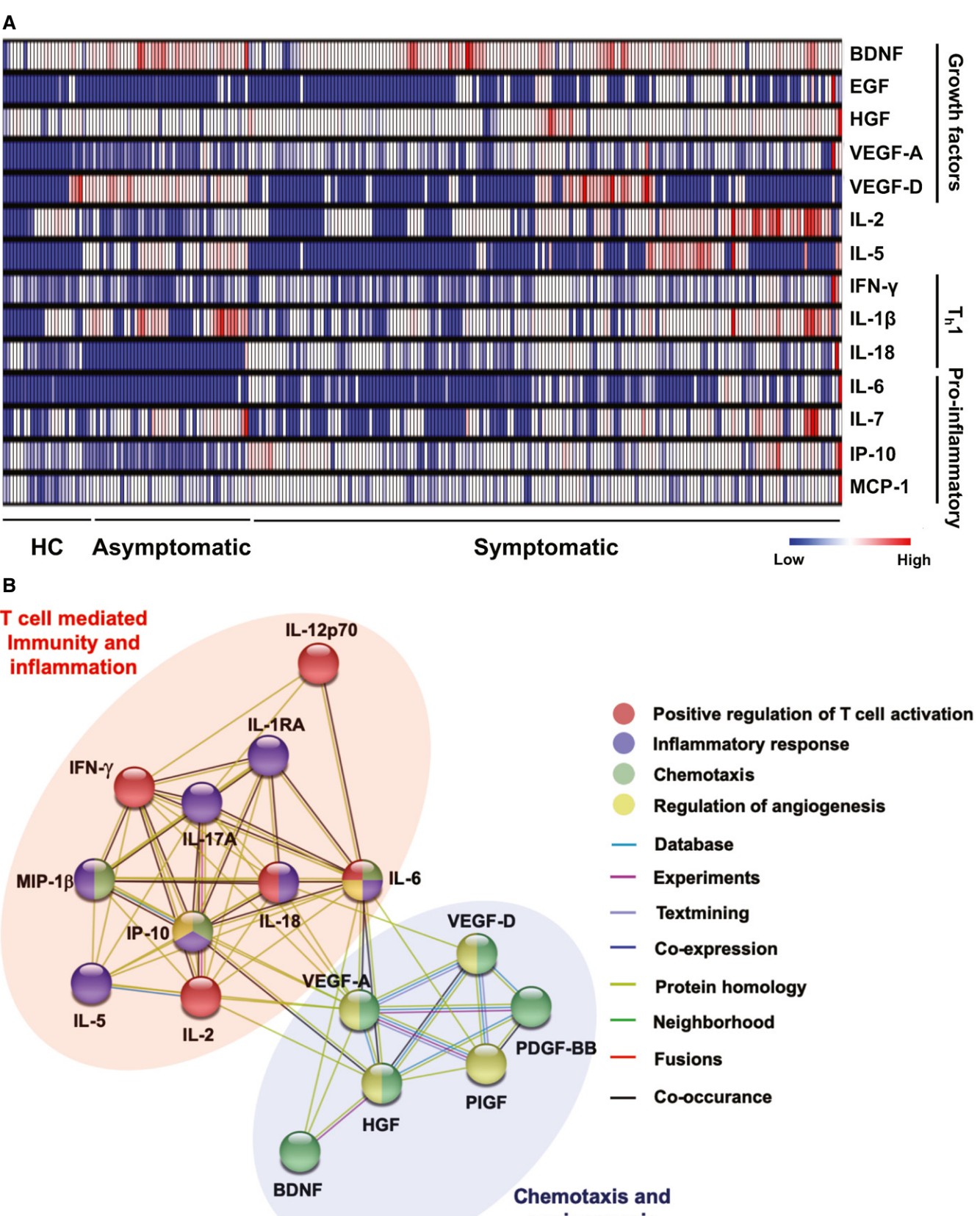

**Figure 2.**

**Figure 2.   Signatures of immune mediators in asymptomatic and symptomatic COVID-19 patients.**

A   Heatmap of immune mediator levels in plasma samples of healthy controls (HC) (*n* = 23) and COVID-19 patients who are asymptomatic (*n* = 48) or symptomatic (*n* = 172). First plasma sample from each SARS-CoV-2 PCR-positive patient was extracted for analyses. Each color represents the relative concentration of a particular analyte (blue = low concentration; red=high concentration). Each row represents one patient. Hierarchical clustering was performed on patients in each severity stratum using MeV software.

B   Network analysis of significant immune mediators between symptomatic and asymptomatic COVID-19 patients. Interactive relationships between the cytokines or chemokines were determined by STRING (Search Tool for the Retrieval of Interacting Genes/Proteins) analysis, with a confidence threshold of 0.5.

Surprisingly, naïve, TEMRA, and central memory (CM) CD4[+] T-cell proportions between symptomatic and asymptomatic patients were similar (Fig 3B and Appendix Fig S1), while asymptomatic patients had a lower proportion of effector memory CD4[+] T cells compared to healthy controls and symptomatic COVID-19 patients. In contrast, the proportions of naïve, TEMRA, and effector memory (EM) CD8[+] T cells were similar between symptomatic and asymptomatic patients, while there were significantly lower CM CD8[+] T cells in asymptomatic compared to symptomatic patients (Fig EV2A). Granzyme expression was comparable between symptomatic and asymptomatic patients (Fig EV2B and C).

In order to assess whether the differences observed at the systemic soluble mediator and transcriptomic levels reflect a functional difference in CD4[+] T cells, a pooled pan-SARS-CoV-2 spike (S), membrane (M), and nucleocapsid (N) peptide re-stimulation as well as PMA/ionomycin re-stimulation of T cells collected shortly after acute infection was performed (median 14.5 days PIO, IQR 14–16.75) (De Biasi *et al*, 2020). Surprisingly, PMA/ionomycin re-stimulated CD4[+] T cells, which are reflective of systemic T-cell responses (Grifoni *et al*, 2020; Le Bert *et al*, 2020), were similar between asymptomatic and symptomatic patients (Fig EV3). Concordantly, asymptomatic patients also had similar levels of SARS-CoV-2 peptide-specific CD4[+] Th1 and Th2 responses as symptomatic patients (Fig 3C). Importantly, asymptomatic patients had more robust SARS-CoV-2-specific Th17 cells than symptomatic patients (Fig 3C), suggesting its potential functional importance in COVID-19 patients.

Despite upregulation of Th2 transcriptomic signatures in asymptomatic patients, which have been shown to mediate humoral response against viral infections (Spellberg & Edwards, 2001), the SARS-CoV-2 spike-specific IgG responses were comparable between the symptomatic and asymptomatic patients (Fig 4A), while spike-specific IgM responses were significantly lower in the asymptomatic patients (Fig 4B). Importantly, the anti-SARS-CoV-2 neutralizing capacity was also markedly lower in asymptomatic patients compared to symptomatic patients against SARS-CoV-2 (Fig 4C and D) (Lee *et al*, 2021). Nevertheless, this suggests that the asymptomatic patients had a markedly distinct T-cell phenotype biased toward the anti-inflammatory profile.

## Inflammatory monocytes and activated neutrophils are markers of pro-inflammatory response

We previously showed that immature neutrophil count was a strong indicator of severity during acute SARS-CoV-2 infection (Carissimo *et al*, 2020). Therefore, we compared the neutrophil profiles between symptomatic and asymptomatic patients using flow cytometry on whole blood (Appendix Fig S2). Asymptomatic patients had higher counts of mature neutrophils compared to symptomatic patients. However, the levels of mature neutrophils in asymptomatic patients were similar to healthy controls. In contrast, both symptomatic and asymptomatic patients had significantly higher levels of immature neutrophils compared to healthy controls (Fig 5A). Next, paired samples from acute and recovered timepoints from asymptomatic patients were assessed to observe longitudinal changes in immune profiles (Fig EV4 and Appendix Figs S3 and S4). We observed no significant variations in immune profiles of asymptomatic patients, suggesting that SARS-CoV-2 infection did not have a drastic impact on the circulating immune cells, unlike previous observations for symptomatic patients (Fig EV4) (Carissimo *et al*, 2020).

To provide a more in-depth view, isolated PBMCs from acute symptomatic and asymptomatic COVID-19 patients were further profiled using CyTOF. Monocytes, which have been described as inflammatory drivers of COVID-19 (Silvin *et al*, 2020), were gated from the CD45[+] leukocyte population (Appendix Fig S5A). Dimensionality reduction method TriMap was performed on the non-natural killer (NK) population to isolate the monocytes from the low-density neutrophils (Appendix Fig S5B). Expression levels of CD14 and CD16, as well as Siglec-1 (CD169) expression, were then used to gate the inflammatory subsets of classical, intermediate, and non-classical monocytes (Fig 5B and C, Appendix Fig S5B and C). We observed that asymptomatic patients had a significantly lower proportion of classical and intermediate monocytes (Fig 5B). Importantly, Siglec-1 expression on these subsets was upregulated in symptomatic patients compared to their asymptomatic counterparts, suggesting less type-I IFN activation in asymptomatic patients (Fig 5C and Appendix Fig S5C) (Bourgoin *et al*, 2020).

**Figure 3.   Asymptomatic COVID-19 patients exhibit more robust SARS-CoV-2-specific T$_h$17 responses compared to symptomatic patients.**

A   Expressions of genes associated with T-cell functionality were compared between acute asymptomatic (*n* = 30) and symptomatic (*n* = 26) COVID-19 patients. Heatmap of DEGs with FDR-adjusted *P* < 0.05, scaled based on log$_2$RPKM values, with blue and red colors indicating low and high expressions, respectively.

B   Mass cytometry was performed on PBMCs obtained from acute symptomatic (*n* = 37) and acute asymptomatic (*n* = 19) COVID-19 patients and healthy controls (*n* = 10). Naïve, TEMRA, central memory (CM), and effector memory (EM) T cells were characterized based on CD45RA and CCR7 expressions. Data are presented as mean ± SD. *\*P* < 0.05 and *\*\*P* < 0.01 (Mann–Whitney *U*-test).

C   SARS-CoV-2-specific CD4[+] non-T follicular helper (TFH) cells from symptomatic (*n* = 5) and asymptomatic COVID-19 (*n* = 5) patients were characterized with flow cytometry based on the expression of IL-17a, IFN-γ, TNF-α, IL-4, and IL-10 upon SARS-CoV-2 peptide stimulation. Representative plot overlay of unstimulated (black) on peptide stimulated (red) was performed on asymptomatic patient. Data are presented as mean ± SD. *\*\*P* < 0.01 (Mann–Whitney *U*-test).

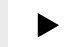

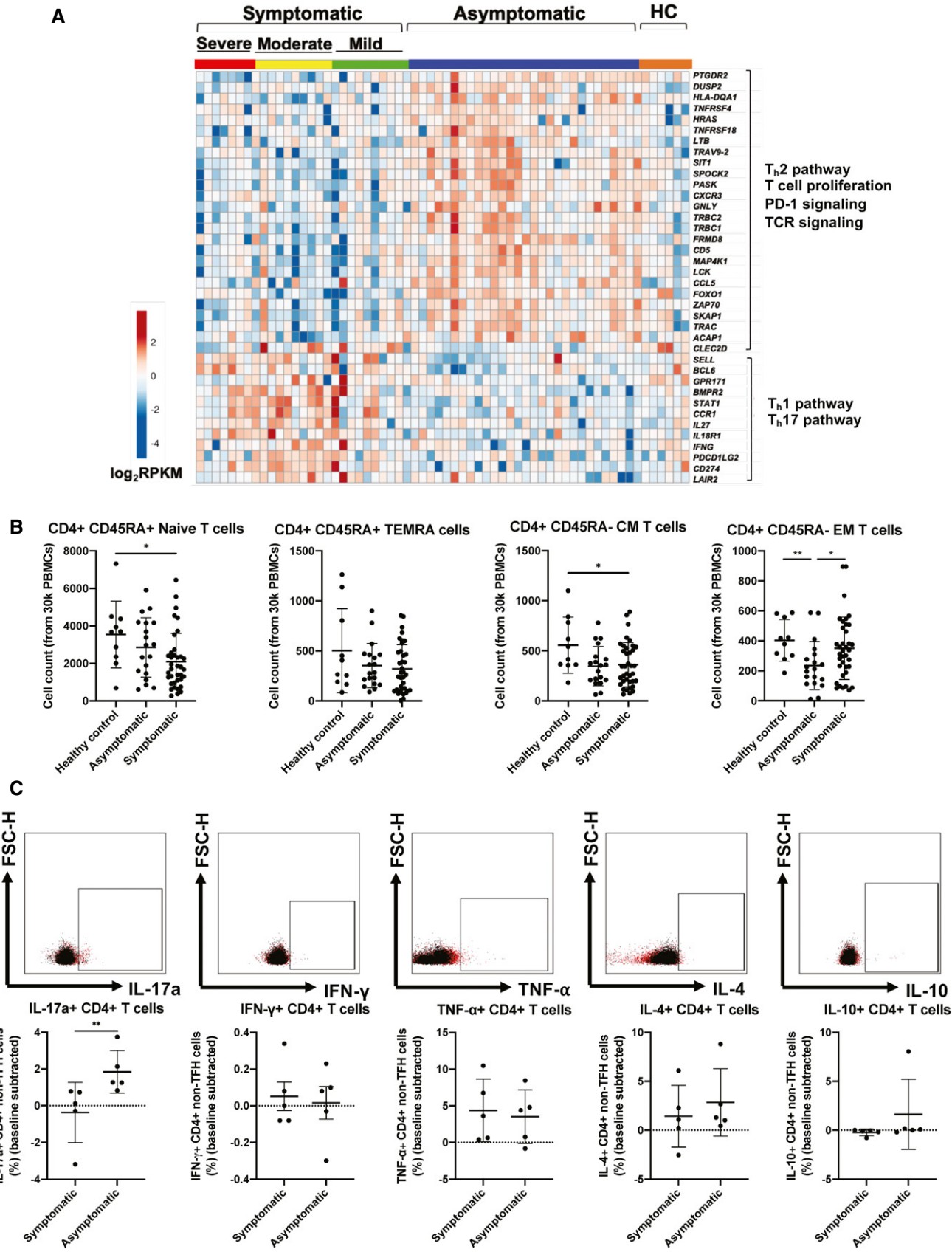

**Figure 3.**

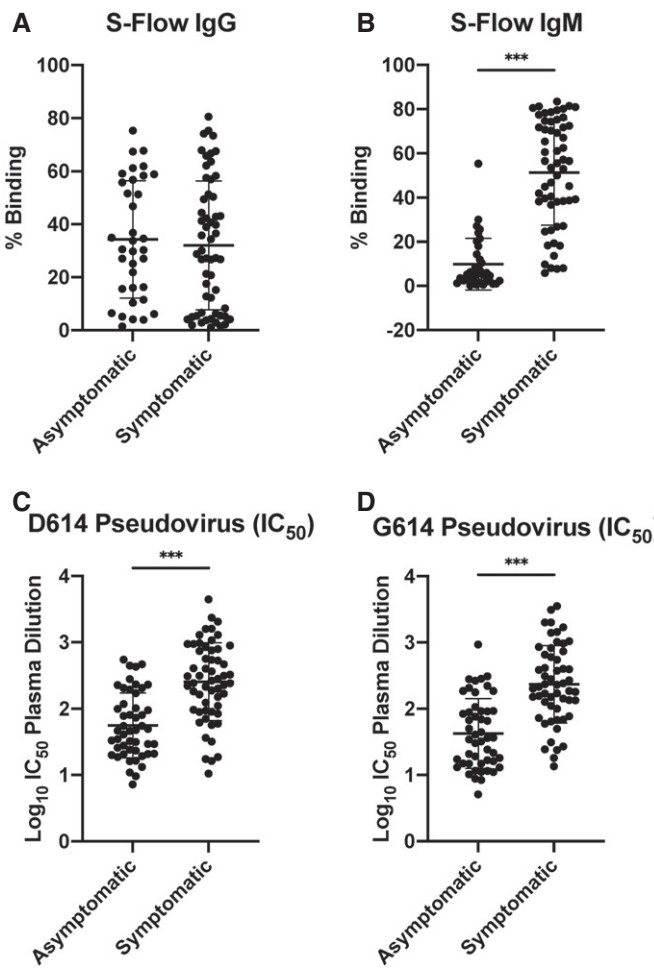

**Figure 4. SARS-CoV-2 spike-specific antibody responses and neutralizing capacity of symptomatic and asymptomatic COVID-19 patients.**

A, B (A) IgG and (B) IgM responses were analyzed by screening plasma samples of asymptomatic ($n = 39$, median study day 29) and symptomatic ($n = 57$, median study day 31) COVID-19 patients. Data are presented as mean ± SD. ***$P < 0.001$ (Mann–Whitney $U$-test).

C, D Plasma samples of asymptomatic (median study day 29) and symptomatic (median study day 31) COVID-19 were assessed for their anti-SARS-CoV-2 neutralization capacity using luciferase expressing lentiviruses pseudotyped with SARS-CoV-2 spike (S) protein of either the original strain, D614, or the mutant variant, G614. Log10 neutralization IC50 profiles against (C) D614 and (D) G614 pseudoviruses at 1 month post-admission (asymptomatic, $n = 49$; symptomatic, $n = 57$ (D614) or 55 (G614)). Data are presented as mean ± SD. ***$P < 0.001$ (Welch's $t$-test).

To confirm these observations, DEGs associated with activated mature neutrophils, inflammatory monocytes and their associated pro-inflammatory mediators were investigated (Fig 5D). In agreement with flow cytometry data, we observed a higher expression of activated neutrophil-associated genes *BST1*, *MXD1*, *FCGR3B*, *STEAP4*, *SELL*, *TLR2*, *VNN3*, and *APOBEC3A* in symptomatic patients. In addition, corroborating the cellular profiles of the patients, increased expression of inflammatory monocyte-associated genes was found in symptomatic patients, including *CLEC7A*, which functions as a pattern recognition receptor and plays a role in innate immune responses (Goodridge *et al*, 2011), and *S100A6*, *S100A12*,

and *S100A8* of the S100 protein family, involved in the regulation of macrophage inflammation (Xia *et al*, 2018; Silvin *et al*, 2020). The expression of pro-inflammatory cytokine and chemokine genes *IL-18*, *TNFSF8*, *TNFSF13B*, *TNFSF4*, *IL27*, *IL1RN*, *TNFSF10*, *IFNG*, *CXCL8*, *CXCL10*, *CCL8*, and *CCL2* was also increased in symptomatic patients (Fig 5D), corroborating the systemic soluble mediator levels, in particular the pro-inflammatory cytokines IL-6 and IL-7 and chemokines IP-10 and MCP-1 (Fig 5E).

## Asymptomatic COVID-19 patients show upregulation of markers associated with cellular repair and leukocyte migration

To identify potential biomarkers that were positively associated with asymptomatic and symptomatic SARS-CoV-2 infection, systemic levels of growth factors were compared between asymptomatic and symptomatic patients (Fig 6A). BDNF, PDGF-BB, and VEGF-D were significantly higher in asymptomatic patients, while the opposite was observed for VEGF-A in symptomatic patients. With this unbalanced expression pattern of the two VEGF isoforms, we hypothesized that the ratio of VEGF-A to VEGF-D could discriminate between asymptomatic and symptomatic COVID-19. Indeed, VEGF-A-to-VEGF-D ratio showed an excellent receiver operating characteristics (ROC) area under the curve (AUC) value of 0.88 for the symptom presence parameter (Fig 6A).

In addition, a large proportion of genes belonging to the family of chemokine receptors (*CXCR3*, *CXCR5*, and *CCR7*) and sphingosine-1 phosphate receptors (*S1PR1*, *S1PR2*, and *SIPR4*) were upregulated in asymptomatic patients compared to symptomatic patients (Fig 6B), suggesting that immune cells in the blood of asymptomatic patients had higher migratory potential. Notably, healthy controls had similar levels of chemokine receptors as asymptomatic patients, suggesting that the migratory potential of immune cells in symptomatic patients might be inhibited. Validating this observation, deep immune cell phenotyping with CyTOF showed that dendritic cells (DCs), B cells, and CD4$^+$ and CD8$^+$ T cells in asymptomatic patients had higher levels of chemokine markers CCR6, CCR7, and CXCR5, highlighting their migratory abilities (Fig 6C).

## Discussion

In order to understand why SARS-CoV-2 infection is asymptomatic in a fraction of patients, we performed a cross-sectional comparison of the transcriptomic signatures, neutralizing capacity of antibodies, cellular phenotypes, and soluble immune profiles of COVID-19 patients from a Singapore cohort. This could help identify immune responses that are associated with pathology or protection against COVID-19. Notably, our cohort contains ethnic differences between asymptomatic and symptomatic patients (Table EV1), which could possibly influence the findings due to existing variance in genetic factors (Sze *et al*, 2020). However, these differences could also be limited as our cohort are of majority Asian ethnicity. Nevertheless, we demonstrated that asymptomatic patients mount a different immune response against SARS-CoV-2 from their symptomatic counterparts, characterized by a less severe myeloid compartment dysregulation, less systemic inflammation, a more robust SARS-CoV-2 peptide-specific Th17 response, and higher levels of tissue healing mediators. While a lower neutralizing antibody capacity

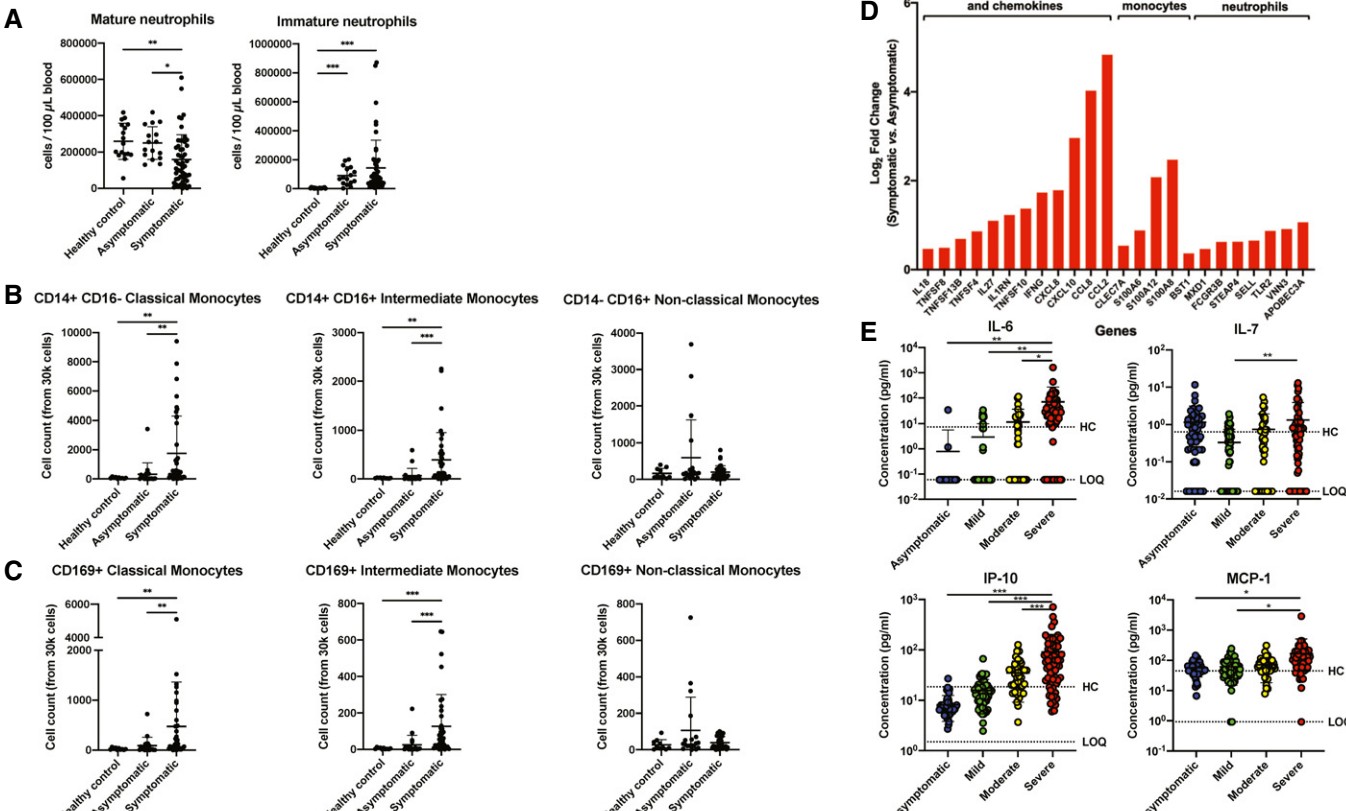

**Figure 5. Asymptomatic COVID-19 patients present less inflammatory monocyte, activated neutrophils, and muted inflammatory response.**

A   Flow cytometry acquisition of patient whole blood was performed, and a number of mature and immature neutrophils were compared between asymptomatic (*n* = 16) and symptomatic (*n* = 52) COVID-19 patients and healthy controls (*n* = 17). Data are presented as mean ± SD. *P < 0.05; **P < 0.01; and ***P < 0.001 (Kruskal–Wallis test with Dunn's multiple comparison).

B   Mass cytometry was performed on PBMCs obtained from acute asymptomatic (*n* = 19) and acute symptomatic (*n* = 37) COVID-19 patients and healthy donors (*n* = 10). Monocytes were characterized based on their CD14 and CD16 expression. Data are presented as mean ± SD. **P < 0.01 and ***P < 0.001 (Kruskal–Wallis test with Dunn's multiple comparison).

C   Classical, intermediate, and non-classical monocytes were further defined in acute asymptomatic (*n* = 19) and acute symptomatic (*n* = 37) COVID-19 patients and healthy donors (*n* = 10) with inflammatory marker CD169. Data are presented as mean ± SD. **P < 0.01 and ***P < 0.001 (Kruskal–Wallis test with Dunn's multiple comparison).

D   Expression data of genes associated with inflammatory monocytes, activated neutrophils, and pro-inflammatory cytokines were compared between acute asymptomatic (*n* = 30) and asymptomatic (*n* = 26) COVID-19 patients and were represented by ratio with respect to expression in asymptomatic patients. Bar graphs show DEGs with FDR-adjusted *P* < 0.05, and fold change values are represented in the log₂ scale.

E   Pro-inflammatory-associated immune mediator levels in plasma fraction samples from first collection timepoint during hospital admission were quantified with 45-plex microbead-based immunoassay. Immune mediator levels were compared between asymptomatic patients (*n* = 48) and symptomatic patients stratified by COVID-19 severity (mild, *n* = 61; moderate, *n* = 43; and severe, *n* = 68). Immune mediator levels for healthy controls (HC) (*n* = 23) are indicated by the black dotted line. Patient samples with concentrations out of measurement range are presented as the value of limit of quantification (LOQ). Data are presented as mean ± SD. *P < 0.05; **P < 0.01; and ***P < 0.001 (one-way ANOVA with post hoc *t*-test).

was observed in asymptomatic patients, they were still able to neutralize virus infection against SARS-CoV-2 wild-type (D614) and the more virulent G614 variants (Lee *et al*, 2021; Plante *et al*, 2021). Importantly, this difference did not impede virus resolution in asymptomatic patients. Our results strongly suggest that asymptomatic patients have developed disease tolerance to immune-mediated pathologies while still developing potent anti-viral immune response, since they were able to clear the virus.

There is accumulating evidence that a dysregulation of myeloid compartment could be a cause of SARS-CoV-2 severity (Silvin *et al*, 2020). Many studies have demonstrated an increase of immature neutrophils and CD169⁺ expressing monocytes (Roussel *et al*, 2021)

in the peripheral blood of severe patients (Carissimo *et al*, 2020), in the in the bronchoalveolar lavage fluid (BALF) (Silvin *et al*, 2020), and in the lungs of severe patients (Barnes *et al*, 2020). The drivers of monocytes/macrophages and neutrophil activation are not yet known. One molecule, calprotectin, produced by activated neutrophils (Silvin *et al*, 2020) can act as a ligand for TLR4 and promote NF-κB activation (Riva *et al*, 2012) and the secretion of multiple COVID-19-associated downstream inflammatory cytokines, i.e., IL-6 and IP-10 (Curtale *et al*, 2013; Young *et al*, 2020). Notably, it has been reported in meta-analysis studies that IL-6 concentrations in COVID-19 patients are significantly lower than patients with ARDS unrelated to COVID-19. This downplays the pro-inflammatory role

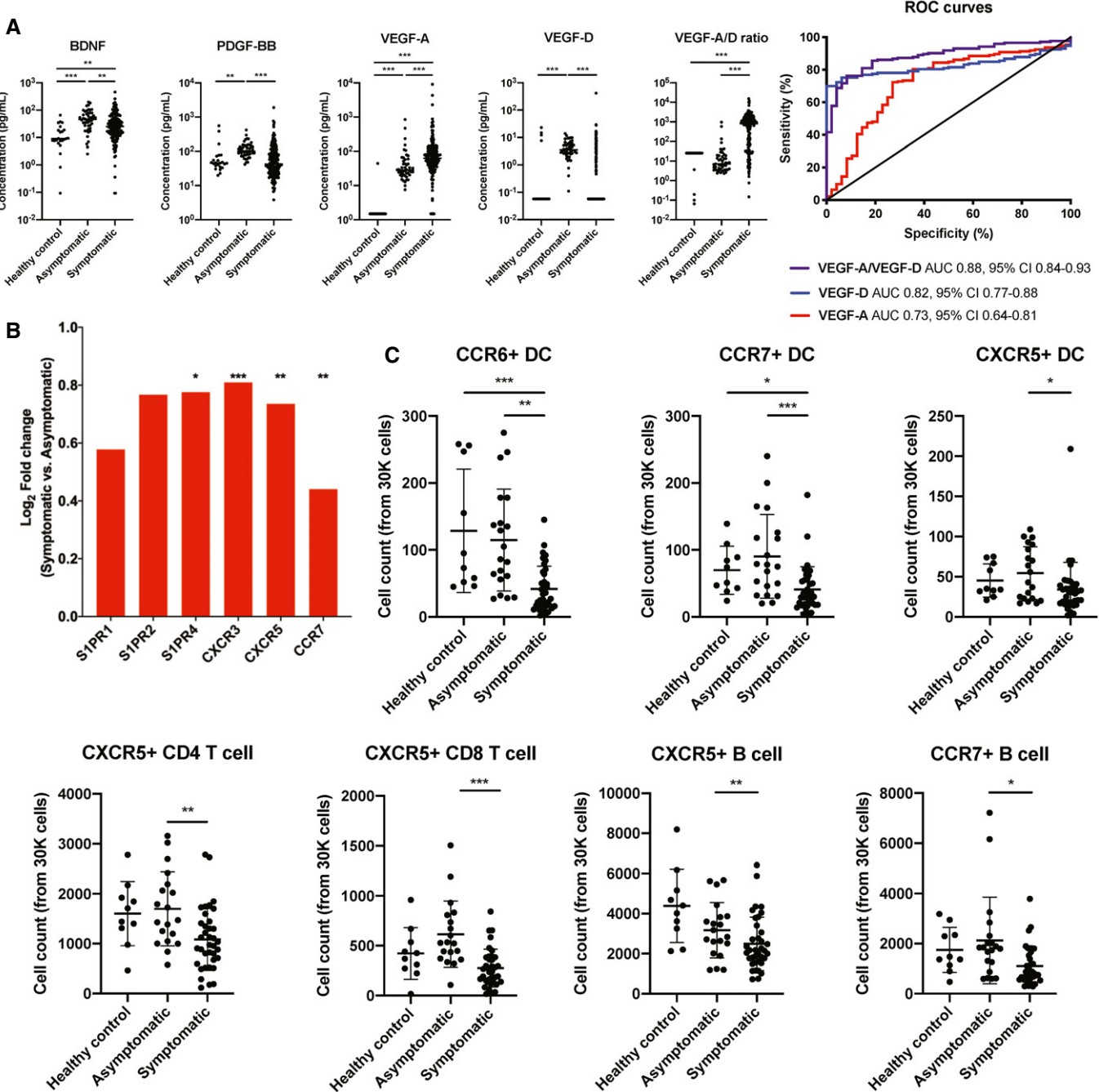

**Figure 6. Asymptomatic patients express higher levels of markers associated with cellular repair and leukocyte migration.**

A   Growth factors in the plasma of asymptomatic ($n = 48$), symptomatic patients ($n = 173$), and healthy controls ($n = 23$) were analyzed by Luminex. A ratio of VEGF-A/VEGF-D was calculated and used to perform receiver operating characteristics analysis. The area under the curve was shown. Data are presented as mean $\pm$ SD. **$P < 0.01$ and ***$P < 0.001$ (Kruskal–Wallis test with Dunn's multiple comparison).

B   RNA-seq analysis was performed on whole blood, and selected transcripts that are over-expressed in asymptomatic patients over symptomatic patients are shown. Genes with false discovery rate less than 0.05 (*), 0.01 (**), or 0.001 (***) are indicated.

C   Cell counts of DCs, CD4 T cells, CD8 T cells, and B cells that express specific chemokine markers between asymptomatic ($n = 19$), symptomatic patients ($n = 37$), and healthy controls ($n = 10$) are analyzed by CyTOF and shown. Data are presented as mean $\pm$ SD. *$P < 0.05$; **$P < 0.01$; and ***$P < 0.001$ (Kruskal–Wallis test with Dunn's multiple comparison).

of IL-6 in severe COVID-19 disease (Leisman *et al*, 2020; Sinha *et al*, 2020). Nevertheless, we showed here that transcriptomic levels of S100A8, which is a subunit of calprotectin, and systemic levels of

pro-inflammatory cytokines in asymptomatic patients were significantly lower than in symptomatic patients, explaining the absence of COVID-19-related symptoms. Our data are in agreement with

another study using a cohort of patients from China (Long *et al*, 2020).

The transcriptome of asymptomatic patients showed a lower level of systemic complement activation compared to symptomatic patients. Systemic complement activation has been associated with respiratory failure in COVID-19 patients (Holter *et al*, 2020) and excessive activation of complement contributes to destructive inflammation harming the host (Ricklin & Lambris, 2013). This may explain the milder inflammation in asymptomatic patients and thereby suggests therapeutic modulation of complement activity at the early phase of disease as an attractive intervention for COVID-19.

In addition to a lower systemic inflammation, asymptomatic patients have a higher systemic levels of PDGF-BB, BDNF, and VEGF-D molecules, which have been associated with endothelial cell repair and angiogenesis (Battegay *et al*, 1994). Their increased production during an active SARS-CoV-2 infection may be responsible for more efficient tissue healing in these patients, thus limiting virus-induced injury to the lungs or other organs. Interestingly, PDGF-BB has also been shown in mouse models to be able to modulate T-cell responses (Daynes *et al*, 1991). It is possible that the higher levels of PDGF-BB in asymptomatic patients are an indicator of a more balanced T-cell response. While we did not observe a difference in Th1/Th2 responses between symptomatic and asymptomatic patients, functional profiling of CD4$^+$ T cells revealed a more robust virus-specific Th17 response in asymptomatic patients. This is consistent with the higher expression of *LCK* and *ZAP70* in asymptomatic patients, which are associated with TCR signaling and T-cell activation (Nika *et al*, 2010; Thill *et al*, 2016). Enhanced TCR signaling may also result in a multitude of cytotoxic T lymphocyte functions (Esser *et al*, 1996). Notably, the immune-regulatory functions of IL-17 produced by Th17 cells, among others, are also associated with anti-viral Th1 cell immunity and cytotoxic T-cell activity, which may enhance virus clearance and hence limit virus-induced damages to the infected host (Bagri *et al*, 2017; Schmidt & Varga, 2018). This is consistent with the upregulation in T-cell cytotoxic activity-related genes observed in asymptomatic patients. Concordantly, *SELL* expression, which is known to be down-regulated upon TCR engagement (preprint: Ivetic *et al*, 2019; Szabo *et al*, 2019), is also found to be lower in asymptomatic patients. A more efficient T-cell anti-viral immunity was postulated in a recent study, which showed a highly functional virus-specific CD4$^+$ T-cell response in asymptomatic infections (Le Bert *et al*, 2021). Taken together, our findings suggest the upregulation of a functionally diverse, virus-specific Th17 response that reinforces anti-viral mechanisms in asymptomatic patients, but yet limits the collateral inflammatory damage as experienced by symptomatic patients.

Our data indicated that immune cells in asymptomatic patients had higher levels of sphingosine-1 receptor and higher levels of chemokine receptors. This finding is consistent with an efficient mobilization of immune cell into secondary lymphoid organs and toward the infected lung (Tiper *et al*, 2016). Interestingly, endothelial sphingosine-1 receptor 1 (*S1PR1*) activation is a key regulator of influenza-virus-induced cytokine storm(Teijaro *et al*, 2011). It will be interesting to assess the expression of *S1PR1* on endothelial cells of the lung to assess its role during the COVID-19 ARDS. This would help assess the feasibility of S1P pathway modulation to help limit the respiratory distress and inflammation in COVID-19 patients, since S1P1R agonist has proven successful to treat mice from ARDS during fatal H1N1 infections (Zhao *et al*, 2019).

Overall, we comprehensively characterized the immune responses in asymptomatic SARS-CoV-2 infection and provided evidence that disease tolerance mechanisms are responsible for prevention/protection of immune-mediated pathologies (Medzhitov *et al*, 2012; Martins *et al*, 2019). While severe symptomatic infections resulted from various levels of immune dysregulation, the asymptomatic patients presented with more robust Th17 cell responses, a sustained neutralizing antibody response and efficient tissue healing process during acute infection. Developing or repurposing therapy that can rectify the immune dysregulation, including limiting inflammation (Kalil *et al*, 2021; Ong *et al*, 2020; RECOVERY Collaborative Group, 2021) or boosting T-cell responses (Le Bert *et al*, 2021), is a viable option to limit COVID-19 progression. We also observed lower EM CD4$^+$ T cells in the peripheral blood of asymptomatic patients during acute infection, while CM CD4$^+$ T cells were proportionally similar in symptomatic patients. Given the long-lasting recall abilities of central memory cells against secondary challenges, long-term protection against COVID-19 should be similar between symptomatic and asymptomatic patients (Roberts *et al*, 2005). Although the manifestation of COVID-19 symptoms was limited in asymptomatic patients through well-balanced innate and adaptive immune responses, they were still infected with SARS-CoV-2 and could be a source of transmission. Forethought and caution should be exercised during mass immunization with novel vaccines, which are still not evaluated on their full efficacy on preventing virus transmission. Nevertheless, our data highlight the importance of balance between anti-viral inflammation, immune-modulation, and tissue repair responses at the early phase of infection, which may aid virus resolution, but yet limit development of symptoms in COVID-19.

# Materials and Methods

### Ethics statement

The study design and protocols for the COVID-19 PROTECT study group were evaluated by National Healthcare Group (NHG) Domain Specific Review Board (DSRB) and approved under study number 2012/00917. Collection of healthy donor samples was approved by SingHealth Centralized Institutional Review Board (CIRB) under study number 2017/2806 and NUS IRB 04-140. Written informed consent was obtained from participants in accordance with the Declaration of Helsinki for Human Research. The experiments conformed to the principles set out in the Department of Health and Human Services Belmont Report.

### Clinical data and biological collection

All individuals with COVID-19 in Singapore are isolated in hospital or dedicated community facilities until deemed to be non-infectious. This includes patients with COVID-19 who are asymptomatic. A total of 263 COVID-19 patients (symptomatic, $n = 215$ and asymptomatic, $n = 48$) were recruited into the study from January to August 2020. Demographic data, disease severity, and clinical laboratory data were obtained from patient records throughout hospitalization or during quarantine (Table EV1). A portion of the symptomatic patients received ongoing treatment during the course

of the study, consisting of anti-virals (lopinavir/ritonavir, hydroxy-chloroquine, remdesivir) or immune-modulators (corticosteroids, interferon, and tocilizumab) (Table EV1). Asymptomatic infections were individuals with a positive SARS-CoV-2 nucleic acid PCR test but without any reported symptoms attributable to COVID-19 in the 3 months before this first test till follow-up 28 days later. All patients included in this study either had a second respiratory sample where SARS-CoV-2 was detected by PCR or positive serology. Blood was collected in VACUETTE EDTA tubes (Greiner Bio, #455036) for healthy donors and acute patients from the day of admission into hospital or in Cell Preparation Tubes (CPT) (BD, #362761) for recovered patients at various timepoints. Whole blood (100µl) was extracted for flow cytometry staining as previously described (Carissimo *et al,* 2020). For all other analyses, blood was collected in Cell Preparation Tubes (CPT) (BD, #362761). Plasma fraction was extracted for multiplex microbead-based immunoassay, and isolated peripheral blood mononuclear cells (PBMCs) were then used for mass cytometry staining after two washes with 1X phosphate buffer saline (PBS). Whole blood samples of COVID-19 patients and healthy controls were also collected into Tempus™ Blood RNA Tubes (Applied Biosystems) and stored at −80°C for transcriptomic profiling.

### Transcriptomic profiling

Tempus™ Blood RNA tubes were heat-inactivated at 60°C for 30 min according to regulatory requirements, followed by RNA extraction using MagMAX™ for Stabilized Blood Tubes RNA Isolation Kit (Invitrogen) as per the manufacturer's instructions. Purified RNA was analyzed on Bioanalyser (Agilent) for quality assessment with RNA integrity number (RIN). Samples with RIN of more than 6 were selected for the study. cDNA libraries were prepared by Smart-Seq v2 (Picelli *et al,* 2014), using a modification of the GlobinLock (GL) method (Krjutškov *et al,* 2016) to block transcription of globin mRNA. The length distribution of the cDNA libraries was monitored using a DNA High Sensitivity Reagent Kit on the LabChip (Perkin Elmer). All samples were subjected to an indexed paired-end sequencing run of $2 \times 151$ cycles on a HiSeq 4000 system (19 samples/lane; Illumina).

STAR aligner (Dobin *et al,* 2013) was used to map paired-end raw reads to human genome build GRCh38 and counted for genes using featureCounts (Liao *et al,* 2014) based on GENCODE v31 gene annotation (Harrow *et al,* 2012). Log$_2$-transformed counts per million mapped read (log$_2$CPM) and log$_2$-transformed reads per kilobase per million mapped reads (log$_2$RPKM) were computed using edgeR Bioconductor package(Robinson *et al,* 2009). Data are accessible at NCBI's Gene Expression Omnibus (GEO) database (GSE155454 and GSE166424). Genes with a log$_2$CPM interquartile range of < 0.5 were removed before subsequent DEG analysis for comparison between symptomatic and asymptomatic profiles was done using edgeR in the R statistical language (version 3.3.3) (R Development Core Team, 2008). Biological processes, canonical pathways, and upstream regulators were predicted from the DEGs with ingenuity pathway analysis (IPA; Qiagen). ClueGO (version 2.5.7), a plug-in app of Cytoscape (version 3.8.0; NIGMS; http://www.cytoscape.org/), was used to better visualize and explore enriched pathways and biological terms related to DEGs. Heatmaps of log$_2$RPKM values for DEGs were generated using ClustVis

(Metsalu & Vilo, 2015), and the rows are clustered using correlation distance and average linkage.

### Multiplex microbead-based immunoassay

Immune mediator levels in the first plasma collected from asymptomatic and symptomatic COVID-19 patients during active infection were quantified by 45-plex microbead-based immunoassays. Plasma samples were subjected to 1% Triton™ X-100 solvent-detergent (SD) mix for virus inactivation prior to quantification. Concentrations of specific analytes were determined with Luminex™ assay, using the Cytokine/Chemokine/Growth Factor 45-plex Human ProcartaPlex™ Panel 1 (Thermo Fisher Scientific). Assay standards and inactivated plasma from COVID-19 patients and healthy controls were incubated with fluorescent-coded magnetic beads pre-coated with respective capture antibodies in a 96-well black clear-bottom plate. After washing, biotinylated detection antibodies were incubated with cytokine-bound beads for 1 h. Finally, streptavidin-PE was added and incubated for another 30 min. Measurements were acquired on the FLEXMAP® 3D (Luminex Corporation, Austin, TX, USA) using xPONENT® 4.0 (Luminex) acquisition software. Data analyses were performed on Bio-Plex Manager™ 6.1.1 (Bio-Rad Laboratories, Hercules, CA, USA). Standard curves were generated with a 5-PL (5-parameter logistic) algorithm, reporting values for both MFI and concentration data.

### Cytometry by time-of-flight (CyTOF) sample processing and data acquisition

Freshly isolated Ficoll-density-centrifuged PBMCs were plated at 0.5–$1 \times 10^6$ in a 96-well V bottom plates and stained for viability with 100 µl of 66 µM of cisplatin (Sigma-Aldrich) for 5 min on ice. Cells were then washed with staining buffer (4% v/v fetal bovine serum, 0.05% v/v sodium azide in 1X PBS) and stained with anti-γδTCR-PE and anti-Vδ1-FITC in 50 µl reaction volume for 15 min at room temperature. Cells were washed with staining buffer and then stained with 50 µl of metal isotope-labeled surface antibodies on ice. After 20 min, cells were washed with staining buffer, followed by PBS, and fixed in 4% v/v paraformaldehyde (PFA, Electron Microscopy Sciences) at 4°C overnight. On the following day, cells were incubated in staining buffer for 5 min. Cellular DNA was labeled at room temperature with 170 nM iridium intercalator (Fluidigm) in 2% v/v PFA/PBS. After 20 min, cells were washed twice with staining buffer.

Prior to CyTOF acquisition, cells were washed twice with water before final re-suspension in water. Cells were enumerated, filtered, and diluted to a final concentration of $0.6 \times 10^6$ cells/ml. EQ Four Element Calibration Beads (Fluidigm) were added to the samples at a final concentration of 2% v/v prior to acquisition. Samples were acquired on a Helios Mass Cytometer (Fluidigm) at an event rate of < 500 events per second. After CyTOF acquisition, data were exported in flow cytometry (FCS) format, normalized and events with parameters having zero values were randomized using a uniform distribution of values between minus-one and zero. Subsequently, manual gating was performed to exclude residual beads, debris, and dead cells.

Triplet-constraint (TriMAP) dimensionality reduction was performed on isolated non-NK cells (Appendix Fig S5B) for the isolation of monocytes from low-density neutrophils (LDN).

Monocytes were identified based on CD14 and CD16 expression, as well as absence of CD24 surface expression.

## Flow cytometry

Whole blood was stained as described previously (Carissimo et al, 2020) with antibodies stated in Appendix Table S1 (100 μl of whole blood per panel) for 20 min in the dark at room temperature (RT). Samples were then supplemented with 1.2× of FACS lysing solution (BD). Final FACS lysing solution concentration taking into account volume in tube before addition is 1×. Samples were vortexed and incubated for 10 min at RT. Subsequently, 500 μl of PBS was added to wash the samples and centrifuged at 300 g for 5 min. Washing step of samples was repeated with 1 ml of PBS. Samples were then transferred to polystyrene FACS tubes containing 10 μl ($1.08 \times 10^4$ beads) of CountBright Absolute Counting Beads (Invitrogen). Samples were then acquired without delay, with vortexing before and every 3 min during acquisition to minimize fixed cell adherence to the tubes, using BD LSRII 5 laser configuration using automatic compensations and running BD FACS Diva software version 8.0.1 (build 2014 07 03 11 47), Firmware version 1.14 (BDLSR II), CST version 3.0.1, and PLA version 2.0. Analysis of flow cytometric data was performed with FlowJo Version 10.6.1. Gating strategies are presented in Appendix Figs S2–S4.

To profile the SARS-CoV-2-specific T effector subsets in the patients, frozen PBMCs from first convalescent timepoint were thawed and rested overnight at 37b0C in RPMI 1640 supplemented with 5% human serum, followed by stimulation with PMA (100 ng/ml, Sigma-Aldrich) and ionomycin (1 μg/ml, Sigma-Aldrich), or pooled SARS-CoV-2 PepTivator S, S1, M and N peptides (0.6 nmol/ml each) (Miltenyi) for 6 h. Brefeldin A and monensin (1×, Thermo Fisher Scientific) were added at 2 h post-stimulation. Cells were stained with surface stain markers in the dark at room temperature for 30 min (Appendix Table S1, intracellular panel no. 1 to 21), followed by fixation and permeabilization for 30 min with Foxp3/Transcription Factor Staining Buffer Set (Thermo Fisher Scientific). Permeabilized cells were then stained for intracellular cytokines for 30 min (Appendix Table S1, intracellular panel no. 22 to 29). Cells were then acquired with the Cytek Aurora cytometer. As the comparison of SARS-CoV-2-specific T-cell responses between symptomatic and asymptomatic patients was retrospective in nature, samples were selected for comparison based on matching study day and sample availability of the PBMCs.

## Anti-SARS-CoV-2 spike protein specific IgG and IgM isotyping

Detection of IgG and IgM specific against the full-length SARS-CoV-2 spike protein was performed using fluorescence-activated cell sorting (FACS) based assay (Goh et al, 2020). Cells expressing full-length SARS-CoV-2 spike protein were seeded at $1.5 \times 10^5$ cells per well in 96-well plates (Thermo Fisher Scientific). The cells were first incubated with plasma samples from COVID-19 patients and healthy controls (1:100 dilution in 10% fetal bovine serum, FBS), followed by a secondary incubation with a double stain, consisting of Alexa Fluor 647-conjugated anti-human IgG or IgM (Thermo Fisher Scientific; 1:500 dilution in 10% FBS) and propidium iodide (Sigma-Aldrich; 1:2,500 dilution). Cells were acquired on a LSRII 4 Laser flow cytometer (BD Biosciences) and analyzed using FlowJo (Tree

Star). A positive antibody response cutoff is defined as mean + 3SD of the healthy controls ($n = 22$).

## SARS-CoV-2 pseudovirus production

The pseudotyped lentiviruses were produced as previously described (Sevajol et al, 2014). Briefly, using the third-generation lentivirus system, pseudotyped viral particles expressing SARS-CoV-2 D614 strain or G614 variant S proteins were generated by reverse transfection of $3 \times 10^7$ of HEK293T cells with 12 μg pMDLg/PRRE (Addgene), 6 μg pRSV-Rev (Addgene), 12 μg pTT5LnX-coV-SP (SARS-CoV-2 wild-type S, a kind gift from Dr. Brendon John Hanson, DSO National Laboratories, Singapore) or pTT5Lnx-coV-SP-D614G (SARS-CoV-2 mutant D614G S), and 24 μg pHIV-Luc-ZsGreen (Addgene) using Lipofectamine 2000 transfection (Invitrogen). Cells were cultured for 3 days, before viral supernatant was harvested by centrifugation to remove cell debris and filtered through a 0.45-μm filter unit (Sartorius). Viral titers were quantified with Lenti-XTM p24 Rapid Titre Kit (Takara Bio).

## Pseudovirus neutralization assay

The pseudotyped lentivirus neutralization assay was performed as previously described, with slight modifications (Sevajol et al, 2014). CHO-ACE2 cells were seeded at $3.2 \times 10^4$ per well in a 96-well black microplate (Corning) in culture medium without Geneticin. Serially diluted heat-inactivated plasma samples (symptomatic, 5-fold dilutions from 1:10 to 1:31,250; asymptomatic, 4-fold dilutions from 1:5 to 1:5,120) were incubated with equal volume of pseudovirus expressing SARS-CoV-2 S proteins of either original wild-type or D614G mutant strain (0.4 ng/μl of p24) at 37°C for 1 h, before being added to pre-seeded CHO-ACE2 cells. Cells were refreshed with culture media after 1-h incubation. After 48 h, cells were washed with PBS and lysed with 1x Passive Lysis Buffer (Promega) with gentle shaking at 125 rpm for 30 min at 37°C. Luciferase activity was subsequently quantified with Luciferase Assay System (Promega) on a GloMax Luminometer (Promega).

## Data processing and statistical analysis

Data analyses were done using GraphPad Prism (GraphPad Software, version 8.4.3). For cytokine analysis, one-way ANOVA with Tukey's multiple comparisons test was used to discern the differences in immune mediator levels among the patients with different clinical severity outcomes. Heatmap and plots were generated using GraphPad Prism version 8. TM4-MeV Suite (version 10.2) was used to compute hierarchical clustering and generate a heatmap of immune mediators, scaling concentrations to between 0 and 1 for visualization. Protein–protein interaction networks of the symptom-associated cytokines were predicted and illustrated with Search Tool for the Retrieval of Interacting Genes/Proteins database (STRING) (version 11.0; available at: https://string-db.org). All the interactions between cytokines were derived from high-throughput laboratory experiments and previous knowledge in curated databases at a confidence threshold of 0.5. For flow cytometry and CyTOF, comparisons of absolute cell count or frequency between healthy control, asymptomatic or symptomatic COVID-19 patients were performed with Kruskal–Wallis test corrected with Dunn's method.

**The paper explained**

**Problem**
The immune responses and mechanisms driving disease progression in symptomatic cases and tolerance in SARS-CoV-2 infection have not been elucidated.

**Results**
Asymptomatic patients had higher counts of mature neutrophils and lower proportion of CD169[+] expressing monocytes in the peripheral blood compared to their symptomatic counterparts. Systemically, levels of pro-inflammatory cytokines were lower in asymptomatic patients, accompanied by higher levels of growth factors and milder pro-inflammatory gene signatures. Mechanistically, a more robust systemic Th2 cell signature with a higher level of virus-specific Th17 cells and a weaker yet sufficient neutralizing antibody profile against SARS-CoV-2 was observed in asymptomatic patients.

**Impact**
Insights gained from this study identify key immune mechanisms underlying disease tolerance, which could serve as therapeutic targets to limit disease progression in COVID-19.

For pseudovirus neutralization assay, comparison of $IC_{50}$ was performed with parametric Welch's *t*-test. All statistical tests were two-sided, and *P*-values < 0.05 were considered to be statistically significant. Exact *P* values are included in Appendix Table S2.

## Data availability

The datasets produced in this study are available in the following databases:

- RNA-seq data: Gene Expression Omnibus GSE155454 https://www.ncbi.nlm.nih.gov/geo/query/acc.cgi?acc = GSE155454
- RNA-seq data: Gene Expression Omnibus GSE166424 https://www.ncbi.nlm.nih.gov/geo/query/acc.cgi?acc = GSE166424

**Expanded View** for this article is available online.

## Acknowledgements

The authors would like to thank all study participants and the healthcare workers caring for COVID-19 patients. The authors also wish to thank the team at NCID, Drs Louis Chai and Gabriel Yan from National University Hospital, Changi General Hospital and Singapore General Hospital for their help in patient recruitment, and Dr. Danielle Anderson and her team at Duke-NUS Medical School for their technical assistance. The authors are also grateful to the staffs from the Singapore Immunology Network (SIgN) Multi-plex Analysis of Proteins (MAP), Immunogenomics (IMG) and Flow Cytometry platforms for their assistance during the study. This work was supported by Singapore Immunology Network (SIgN) core research grant and the A*STAR COVID-19 Research funding (H/20/04/g1/006) provided to SIgN by the Biomedical Research Council (BMRC). Subject recruitment, sample collection, and analyses were funded by the National Medical Research Council (NMRC) COVID-19 Research Fund (COVID19RF-001, COVID19-RF007, COVID190RF-060). The SIgN Immunomonitoring Platform is supported by a BMRC IAF 311006 grant and BMRC transition funds #H16/99/b0/011. The SIgN Flow

Cytometry and the Multiple Analyte Platforms were supported by a grant from the National Research Foundation, Immunomonitoring Service Plat-form ISP) (#NRF2017_SISFP09) and the National Research Foundation Singapore (NRF). A Torres-Ruesta is supported by the A*STAR Singapore International Graduate Award (SINGA) scholarship. The funders had no role in the design and conduct of the study; collection, management, analysis, and interpretation of the data; preparation, review, or approval of the manuscript; and decision to submit the manuscript for publication. The corresponding author had full access to all the data in the study and takes responsibility for the integrity of the data and the accuracy of the data analysis.

## Author contributions

Y-HC, S-WF, C-MP, GC, and NK-WY conceptualized, processed, acquired, analyzed, and interpreted the data and wrote the manuscript. SNA, RS-LC, AT-R, CY-PL, MZRT, and ZWC processed, acquired, and analyzed the data. YSG performed the anti-SARS-CoV-2 spike protein IgG and IgM isotyping. JWL and WLX designed the immunophenotyping panels and analyzed the data. BL analyzed and interpreted the transcriptomic data. W-HL, BW, and CW performed the pseudovirus neutralization assay. S-YT, SK, BEY, Y-SL, and DCL designed and supervised sample collection. OR, LR, and LFPN conceptualized, designed, analyzed, and wrote the manuscript. All authors revised and approved the final version of the manuscript.

## Conflict of interest

The authors declare that they have no conflict of interest.

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
