## [Review Process File · EMBO Molecular Medicine]

Asymptomatic COVID-19: disease tolerance with efficient anti-viral immunity against SARS-CoV-2

Yi-Hao Chan, Siew-Wai Fong, Chek Meng Poh, Guillaume Carissimo, Nicholas Kim-Wah Yeo, Siti Naqiah Amrun, Yun Shan Goh, Jackwee Lim, Weili Xu, Rhonda Sin-Ling Chee, Anthony Torres-Ruesta, Cheryl Yi Pin Lee, Matthew Zirui Tay, Zi Wei Chang, Wen-Hsin Lee, Bei Wang, Seow-Yen Tan, Shirin Kalimuddin, Barnaby Young, Yee-Sin Leo, Cheng-I Wang, Bennett Lee, Olaf Rotzschke, David Lye, Laurent Renia, and Lisa Ng

DOI: 10.15252/emmm.202114045

Corresponding author(s): Lisa Ng (lisa_ng@immunol.a-star.edu.sg) , Laurent Renia (renia_laurent@IDLabs.a-star.edu.sg)

Review Timeline:

Submission Date:	29th Jan 21
Editorial Decision:	19th Feb 21
Revision Received:	6th Apr 21
Editorial Decision:	23rd Apr 21
Revision Received:	26th Apr 21
Accepted:	28th Apr 21

Editor: Zeljko Durdevic

Transaction Report:

19th Feb 2021

Dear Prof. Ng,

Thank you for the submission of your manuscript to EMBO Molecular Medicine. We have now received feedback from the three reviewers who agreed to evaluate your manuscript. As you will see from the reports below, the referees acknowledge the interest of the study but also raise serious concerns that should be addressed in a major revision.

Addressing the reviewers' concerns in full will be necessary for further considering the manuscript in our journal, and acceptance of the manuscript will entail a second round of review. EMBO Molecular Medicine encourages a single round of revision only and therefore, acceptance or rejection of the manuscript will depend on the completeness of your responses included in the next, final version of the manuscript. For this reason, and to save you from any frustrations in the end, I would strongly advise against returning an incomplete revision.

We realize that the current situation is exceptional on the account of the COVID-19/SARS-CoV-2 pandemic. Therefore, please let us know if you need more than three months to revise the manuscript.

I look forward to receiving your revised manuscript.

Yours sincerely,

Zeljko Durdevic

***** Reviewer's comments *****

Referee #1 (Comments on Novelty/Model System for Author):

the consortium performing the study has an outstanding competence on all modern methods used in this investigation. The presented data and the suppl material are very relevant for most of them. The ability to obtain molecular data in asymptomatic patients is of major interest to understand the evolution towards severe forms. The results are very informative.

Referee #1 (Remarks for Author):

The manuscript "Asymptomatic COVID-19: disease tolerance with efficient anti-viral immunity

against SARS-CoV-2" addressed a fundamental question about the immune response and the underlying mechanisms driving the progression from asymptomatic to symptomatic status after infection by the SARS-CoV-2. The methods used to decipher these aspects included the transcriptomic profiles, cytokine responses, neutralization capacity of antibodies and cellular immune phenotype of asymptomatic patients in comparison with symptomatic patients. This study concluded that "asymptomatic patients mount less pro-inflammatory and more protective immune responses against SARS-CoV-2 indicative of disease tolerance." This "study highlight key immune pathways that could serve as therapeutic targets to prevent disease progression in COVID-19".

Major comments

The focus of the paper is essential and may provide important knowledge for fighting the COVID-19 outbreak both with immune therapies and/or vaccine protection. Works on basic aspects for immune response of asymptomatic patients has been scarcely reported, especially for cellular response. The study showed a balanced cellular immune response, which is pro-inflammatory et more protective potentially related an acquired tolerance.

The authors have to be congratulated for this translational work including a large cohort of patients.

First, could the authors mention the treatment received by the different cohorts of patients, particularly those that may influence the immune response, as steroids or anti-IL-6...

Second, the timing at which the samples have been performed could be clarified: for the asymptomatic 3 days post-admission: what was the reason for admission? For the symptomatic patients between day 4 and 11 post illness onset. Do the authors think that it could change rapidly along time evolution? If yes, please recognize that it could be a limitation. Please mention it.

Transcriptomic data: could the authors explain why or how the patient selection for this part of the study was made? It is unclear for the reviewer looking at the fig 1 A and B, finally what type of concerned genes are different between asymptomatic and symptomatic, even the PCA showed clear differences. Could the under and upper-regulated genes be categorized in symptomatic patients in additional data? Fig 1C: is the description by under or over-expressed the correct way to illustrate: does it relate to amount of expression in asymptomatic referring to the results of symptomatic patients? Are the down and up regulated genes in a lesser amplitude in the 2groups? Please clarify.

Systemic inflammation: do the authors think that measuring systemic markers of inflammation may reflect well what happens in the lung, or simply the systemic inflammatory response stimulated by different mechanisms? The figure 2A shows within the symptomatic patients various response for IL-5 from deep downregulation to moderate upregulation. Knowing the major role of this cytokine for the inflammatory response of adipose tissue, did these differences fit well with clinical markers of obesity? This aspect could be more discussed in the discussion, considering the high BMI as a risk factor for severe forms of COVID-19. The results for IL-6 levels in the heatmap are important, since some symptomatic patients had a low level as observed for asymptomatic. Does it question the role of IL-6 in this disease? Please comment.

TH17 response: figure 3A depicts well the high expression of the TH2 genes associated with a moderately high expression of the TH1 and TH17. The reviewer is not sure that the 3B and C are useful for the main draft and could be moved to additional data.

Monocytes and neutrophils: the presented data are very convincing about the differences within monocyte subclasses and their sensitivity to type I IFN. Fig 4D: it would be informative to describe in the legend, how the gene expression in different cell subpopulations were compared between asymptomatic vs symptomatic patients: subtraction, ratio, please clarify. Fig 4E: why the systemic mediators in patients were not compared to healthy controls? Looking at the IL-6 values, it is

remarkable to observe an increase in patients, which are nevertheless largely lower than the levels reported in other causes of acute respiratory failure with ARDS, which can also be discussed in the Discussion section.

Discussion:

The discussion is well conducted and informative. Some aspect might be seen from different point of view. P14 last {section sign} discussed the calprotectin level lowered in asymptomatic vs the symptomatic patients. If the hypothesis is correct, it can also be seen as DAMPs molecule related to the cell damage as observed in severe septic patients, maintaining a high plasma level. Were differences between the severity (mild, moderate, severe) patients existed?

P15 {section sign} 2 appears mainly hypothetic and questions the interest to look only at the blood to study the immune response in asymptomatic patients. Even ethically difficult to justify, information from BALavage would be more than important to support the hypothesis. L311 to 318 appears also elusive and could be rewritten.

P 16 L3275 mentioned a "cytokine storm" that is not demonstrated in severe COVID patients as observed in septic or ARDS patients as recently shown. Consequently, it could be then wrong to consider the cytokine storm as a target for immune-modulation drug, as tocilizumab.

Referee #2 (Remarks for Author):

In the manuscript entitled "Asymptomatic COVID-19: disease tolerance with efficient anti-viral immunity against SARS-CoV-2" by Chan and collaborators, the authors aimed to investigate the immune response profiles underlying asymptomatic SARS-CoV-2 infection. The study was performed on patients recruited for a COVID-19 PROTECT cohort in Singapore. Subjects were classified as symptomatic (n=215) and asymptomatic (n=48), based on a positive SARS-CoV-2 PCR test with or without reported COVID-19 symptoms, respectively. Whole blood was collected and PBMC/plasma used for transcriptomic profiling using the HiSeq 4000 System Illumina technology, multiplex microbead-based immunoassay, CyTOF and Flow cytometry analysis, and antibody-mediated pseudovirus neutralization assay. The results support the idea that asymptomatic patients display a protective moderate pro-inflammatory adaptive cellular immune response. The authors concluded that asymptomatic patients exhibit disease tolerance.

Results in Figure 1 depict transcriptomic signatures of symptomatic and asymptomatic patients revealed by high throughput RNA sequencing. Differentially expressed genes (DEG) were identified. Principal component analysis (PCA) show a clear segregation between symptomatic and asymptomatic patients. Gene ontology (GO) analysis revealed the biological functions of DEGs over and under expressed in asymptomatic patients. Results in Figure 2 depict the expression of plasma markers quantified by Luminex immunoassay, with a focus on differentially expressed growth factors and pro-inflammatory cytokines. Asymptomatic individuals showed higher levels of certain growth factors (e.g., BDNF) and lower levels of pro-inflammatory mediators. Results in Figure 3 depict T cell-associated DEG, as well as CyTOF and flow cytometry validations of T cell subsets (Naives, TEMRA, CM and EM) and Th1/Th17 profiles. Of note, the % of SARS-CoV-2-specific T cells producing IL-17A was higher in asymptomatic compared to symptomatic patients. In Figure 4 mature/immature neutrophils were quantified by flow cytometry, monocyte subsets were analysed by CyTOF, and Luminex immunoassay was used to quantify cytokines/chemokines. In the Figure 5 results show that symptomatic compared to asymptomatic patients display decreased expression of markers associated with cellular repair and leukocyte migration, such as CCR6, CCR7 and CXCR5 on DC, T cells and B cells. Finally, Figure 6 summarizes the main findings of the study.

Briefly, the authors propose a model in which symptomatic infection is a result of immune dysregulation, while asymptomatic patients mount a strong Th17 cell response, a sustained neutralizing antibody activity and effective tissue healing process during acute infection.

The research topic is highly relevant for the current SARS-CoV-2 pandemic. The paper is straightforward and overall easy to read. The study idea of assessing early acute immune responses response is very interesting. Diverse techniques were employed that showed coherent results. However, multiple aspects need to be addressed prior publication, as listed below:

Major criticisms:

1. The results on Ab-mediated neutralisation (Figure EV2) should be presented as a main figure. The authors should also include data on the quantification of these Abs in the plasma (IgG and IgM).
2. The authors mention differences in CT values between symptomatic and asymptomatic patients (lines 120-123). However, in Table EV1 these differences were not so dramatic (30 vs 32). What are numbers between brackets? Please indicate the limit of detection and the CT value of the negative control. Also, was plasma viremia measured in the two groups?
3. It will be important to provide a brief definition of the term tolerance, as to avoid inaccurate interpretations. One could interpret tolerance as absence of immune response.
4. For all figures showing gene expression the contrast should be clarified and should be the identical throughout the manuscript (e.g., asymptomatic versus symptomatic). Figure legends should clarify color codes used.
5. Figure 1: the authors should provide a study design and better explain the inclusion criteria of asymptomatic and symptomatic patients at different times post-admission and post-illness onset. Such differences may account for differences in the results.
6. Figure 1C: the complement activation genes are under expressed. This is an important finding not discussed in the manuscript. The authors should discuss how early complement activation monitoring can be used to identify patients at risks of developing severe disease in order to treat them early. The same observation for responses to type I IFN. Could the authors measure plasma or cell associated IFN? Can complement and IFN be better predictors for disease aggravation?
7. Figure 2A: Is it possible to add healthy control data set in the heatmap?
8. Figure EV1 legend, the authors should clarify/rewrite this sentence "First plasma sample from each SARS-CoV-2 PCR-positive patient was extracted for analyses"
9. Figure 2A, it is unclear why IL-7 is considered pro-inflammatory?
10. Lines 179-180: it is unclear how DEGs were classified in Th1, Th2, and Th17
11. Figure 3A: please clarify how the stratification in Th subsets was performed. Curiously, Th2 cells appear to express more Lck and ZAP-70, a characteristic of Th17 cells. Also, SELL or L-Selectin is lost with cell activation; or here it is expressed at the highest levels in symptomatic patients. Please discuss these findings.
12. Figure 3: It is surprising that in asymptomatic patients, effector memory CD4 T cells decreased compared to symptomatic but in DEG analysis there was an upregulation of T cell proliferation. Authors need to justify this result. Also, authors should graphically represent the data for both study groups for Ki67 and activation marker for both CD4 and CD8 subsets. It will be important to understand how cell proliferation and activation state is changing with disease severity.
13. Figure 3C: Although the idea that asymptomatic compared to symptomatic patients develop more robust SARS-CoV-2-specific Th17 responses is interesting, the number of participants in these comparisons is too low to allow conclusions. The authors should provide sample size calculations for comparisons between groups. Does plasma IL-17A levels also vary in the same direction?
14. Are there other clinical parameters available, such as C Reactive Protein and d-Dimer levels, to

allow a better correlation between study findings and pre-existing conditions?

15. Discussion: One interesting finding is BDNF. Can a decrease in BDNF explain the neuropathogenesis of SARS-CoV-2 in symptomatic patients?

16. Discussion: the application of this study is not only to characterize asymptomatic patients but to identify biomarkers that can as early as possible predict severe disease evolution in symptomatic patients. Such predictors should be discussed (complement, IFN).

17. Discussion: Authors could further discuss the impact of the study's result, on how it could be used to find therapeutic targets to fight COVID-19.

Minor criticisms:

1. Although in lines 115-120 the authors describe the ethnicity of the participants, in Table E1, authors need to include other Ethnicities, not only Chinese.

2. Please write ThX and not ThX

3. On the line 69, to clarify what is together: the data or the patients?

4. On the line 272, the sentence should be reorganized since it the word transcriptomic looks like it's detached.

5. On the line 296: "a another" English mistake.

6. On the line 332: "pathologie" English mistake.

Referee #3 (Comments on Novelty/Model System for Author):

There have already been a few papers on the immune profiles of asymptomatic COVID-19 patients, though this study is likely the most comprehensive. The study is confounded by differences in the two groups (ethnicity, timing of sample collection, etc) that limit some of the findings.

Referee #3 (Remarks for Author):

Chan et al. present a detailed immunologic study of asymptomatic COVID-19 subjects compared to symptomatic subjects. They obtained peripheral blood at convalescent timepoints and use a variety of complementary techniques to characterize the overall host response (analysis of blood transcriptomes, Luminex of serum cytokines and growth factors, mass and flow cytometry) as well as the anti-viral response (neutralization assays, intracellular cytokine staining after peptide pool stimulation). The main findings are that "asymptomatic patients mount less pro-inflammatory and more protective immune responses against SARS-CoV-2 indicative of disease tolerance." Overall, I found the presentation of data in main and supplemental figures to be complete and thorough and the paper easy to read. The methods are thoroughly described and statistical analyses are mostly justified. To my knowledge, the paper represents one of the more thorough immunologic analyses of asymptomatic COVID-19 patients. My suggestions are mostly focused on interpretation of the findings and some minor clarifications.

A significant limitation in the study design is that a minority of the asymptomatic subjects were ethnic Chinese and likely had different exposures by virtue of their lifestyles and profession. Thus the results may be confounded by both genetics and exposure history. This should be discussed as a limitation, especially in the context of emerging data that genetics plays a role in COVID-19 outcomes (including GWAS studies). Another source of confounding is the timing of sample collection, which was uniformly earlier among asymptomatic subjects. This should also be discussed

as a limitation.

The Discussion seeks to claim that the data explain "why SARS-CoV-2 infections are asymptomatic" (Line 271). But the study is cross-sectional, not longitudinal, and uses samples collected after symptom onset, not before. Thus, any language about causal inference is an over interpretation of the data presented. Rather, the authors have only shown associations (albeit important ones), and this should be clearly acknowledged throughout the text. On that note, Figure 6 is purely speculative and should be removed entirely.

Minor

It is unclear to me why asymptomatic patients were admitted to the hospital.

Fig EV1. IL-1 β and IL-18 are not Th1 cytokines. In fact, it is much more likely that they were produced by myeloid cells prior to detection in plasma.

Fig EV3. It is inaccurate to define CD4+CD25+ T cells as Tregs. CD25 is expressed on activated T cells, and it is well known that this definition does not fully correlate with FoxP3 staining.

Fig EV3. Gating for Granzyme B is not shown.

Table EV3 lists 29 markers as part of a flow cytometry panel that was run on an LSRII cytometer (Line 456). This is technically impossible, I believe. Was the instrument a BD Symphony or Aurora Cytex?

Point-by-point response to reviewers' comments EMM Submission EMM-2021-14045

***** Reviewer's comments *****

Referee #1 (Comments on Novelty/Model System for Author):

the consortium performing the study has an outstanding competence on all modern methods used in this investigation. The presented data and the suppl material are very relevant for most of them. The ability to obtain molecular data in asymptomatic patients is of major interest to understand the evolution towards severe forms. The results are very informative.

Response: We thank the reviewers for the positive comments.

Referee #1 (Remarks for Author):

The manuscript "Asymptomatic COVID-19: disease tolerance with efficient anti-viral immunity against SARS-CoV-2" addressed a fundamental question about the immune response and the underlying mechanisms driving the progression from asymptomatic to symptomatic status after infection by the SARS-CoV-2. The methods used to decipher these aspects included the transcriptomic profiles, cytokine responses, neutralization capacity of antibodies and cellular immune phenotype of asymptomatic patients in comparison with symptomatic patients. This study concluded that "asymptomatic patients mount less pro-inflammatory and more protective immune responses against SARS-CoV-2 indicative of disease tolerance." This "study highlight key immune pathways that could serve as therapeutic targets to prevent disease progression in COVID-19".

Major comments

The focus of the paper is essential and may provide important knowledge for fighting the COVID-19 outbreak both with immune therapies and/or vaccine protection. Works on basic aspects for immune response of asymptomatic patients has been scarcely reported, especially for cellular response. The study showed a balanced cellular immune response, which is pro-inflammatory et more protective potentially related an acquired tolerance. The authors have to be congratulated for this translational work including a large cohort of patients.

Response: We thank the reviewer for the positive comments.

First, could the authors mention the treatment received by the different cohorts of patients, particularly those that may influence the immune response, as steroids or anti-IL-6...

Response: A portion of symptomatic patients were on anti-virals (Lopinavir/ ritonavir, Hydroxychloroquine, Remdesivir) or Immunomodulators (Corticosteroids, Interferon, and Tocilizumab). Some patients received more than one agent. The number of patients with ongoing drug treatment during the study has now been included in Table EV1 and mentioned in the material and methods section, on page 19, lines 399 to 402, to read "A portion of the symptomatic patients received ongoing treatment during the course of the study, consisting of anti-virals (Lopinavir/ ritonavir, Hydroxychloroquine, Remdesivir) or immune-modulators (Corticosteroids, Interferon, and Tocilizumab) (Table EV1)."

Second, the timing at which the samples have been performed could be clarified: for the asymptomatic 3 days post-admission: what was the reason for admission?

Response: In Singapore, all individuals with COVID-19 are isolated in hospitals or dedicated community facilities until deemed to be non-infectious (PCR negative). This includes patients with COVID-19 who are asymptomatic, where the infection was detected during routine screening. Patient admission date is defined as the day they were admitted into these facilities. Patients were still PCR positive for SARS-CoV-2 during admission. The text has now been edited on page 19, lines 397 to 399 to read “Demographic data, disease severity and clinical laboratory data were obtained from patient records throughout hospitalization or during quarantine (Table EV1)”.

For the symptomatic patients between day 4 and 11 post illness onset. Do the authors think that it could change rapidly along time evolution? If yes, please recognize that it could be a limitation. Please mention it.

Response: We recruited the patients when they were still symptomatic and PCR positive. We do recognize that timing of blood sample collection, performed on 4 to 11 days post illness onset, could result in differences in cytokine/chemokine levels in the symptomatic patients. However, this is a limitation that cannot be avoided due to the different timing of diagnosis and/or hospitalization of the patients. Therefore, we have added a sentence in the results section on page 7, line 135 to 137, to reflect this. The sentence reads “Due to the differences in timing of diagnosis and hospitalization, blood sample collections were performed at varying stages of the acute phase, which could influence the observed transcriptomic profiles in symptomatic patients.”

Transcriptomic data: could the authors explain why or how the patient selection for this part of the study was made? It is unclear for the reviewer looking at the fig 1 A and B,

Response: We wish to explain that the patients were selected for transcriptomic analysis based on their sample availability and sample RNA quality. Only acute samples from symptomatic and asymptomatic patients with RNA integrity number (RIN) of more than 6 were included in this study. This is described in the results section on page 7, lines 126 to 128, and in the methods section on page 20, line 422, which reads “We studied 48 asymptomatic COVID-19 patients and compared their transcriptomic signatures, soluble immune mediator levels and immune cell profiles during acute infection against 172 symptomatic patients (Table EV2).” and “Samples with RIN of more than 6 were selected for the study.”. Additional description on the selection criteria has now been included in the legend of Figure 1 to read “Only samples with RNA integrity number >6 were sent for sequencing and included in the analysis”.

Finally, what type of concerned genes are different between asymptomatic and symptomatic, even the PCA showed clear differences. Could the under and upper-regulated genes be categorized in symptomatic patients in additional data? Fig 1C: is the description by under or over-expressed the correct way to illustrate: does it relate to amount of expression in asymptomatic referring to the results of symptomatic patients? Are the down and up regulated genes in a lesser amplitude in the 2groups? Please clarify.

Response: The under-expressed and over-expressed genes are indicated in Dataset EV1, where a negative fold change indicates lower gene expression in asymptomatic patients compared to symptomatic patients, while a positive fold change indicates higher gene expression in asymptomatic patients compared to symptomatic patients. Additional description has been added into the legend of Figure 1C to read “Illustrated genes were under-expressed or over-expressed in asymptomatic patients compared to symptomatic patients.”.

Systemic inflammation: do the authors think that measuring systemic markers of inflammation may reflect well what happens in the lung, or simply the systemic inflammatory response stimulated by different mechanisms?

Response: We do acknowledge that there may be some deviation due to tissue-specific immunity (Dorward et al, 2021). However, due to sample availability, we quantified systemic markers of inflammation for a reflection of immune response differences between symptomatic and asymptomatic patients.

Citation:

1. Dorward, D. A., Russell, C. D., Um, I. H., Elshani, M., Armstrong, S. D., Penrice-Randal, R., ... & Lucas, C. D. (2021). Tissue-specific immunopathology in fatal COVID-19. *American journal of respiratory and critical care medicine*, 203(2), 192-201.

The figure 2A shows within the symptomatic patients various response for IL-5 from deep downregulation to moderate upregulation. Knowing the major role of this cytokine for the inflammatory response of adipose tissue, did these differences fit well with clinical markers of obesity? This aspect could be more discussed in the discussion, considering the high BMI as a risk factor for severe forms of COVID-19.

Response: We thank the reviewer for this comment. We performed the correlation of IL-5 with BMI for the symptomatic patients. Results are attached below for review purposes. We did not observe a tight correlation of IL-5 and BMI, hence we do not think that obesity was a significant risk factor in our cohort that contributed to the severe forms of COVID-19.

The results for IL-6 levels in the heatmap are important, since some symptomatic patients had a low level as observed for asymptomatic. Does it question the role of IL-6 in this disease? Please comment.

Response: We thank the reviewer for bringing up this important note on IL-6. Although some symptomatic patients had low levels of IL-6 in our study, this does not downplay its pathogenic role. We wish to note that symptomatic patients lay across a severity spectrum. IL-6 levels have been shown to be associated with disease severity in previous studies (Young et al, 2020, Huang et al, 2020), with lower levels of IL-6 observed in symptomatic patients with milder disease. This may explain why IL-6 are observed to be low in some patients.

Citation:

1. Young, B. E., Ong, S. W., Ng, L. F., Anderson, D. E., Chia, W. N., Chia, P. Y., ... & Singapore 2019 Novel Coronavirus Outbreak Research Team. (2020). Viral Dynamics and Immune Correlates of Coronavirus Disease 2019 (COVID-19) Severity. *Clinical Infectious Diseases*.
2. Huang, C., Wang, Y., Li, X., Ren, L., Zhao, J., Hu, Y., ... & Cao, B. (2020). Clinical features of patients infected with 2019 novel coronavirus in Wuhan, China. *The lancet*, 395(10223), 497-506.

TH17 response: figure 3A depicts well the high expression of the TH2 genes associated with a moderately high expression of the TH1 and TH17. The reviewer is not sure that the 3B and C are useful for the main draft and could be moved to additional data.

Response: We wish to explain that while Figure 3A depicts the transcriptomic signature differences between symptomatic and asymptomatic patients, Figure 3B provides the activation and differentiation status of CD4 T cells by mass spectrometry, while Figure 3C profiles the cytokine expression of CD4 T cells in COVID-19 patients during re-stimulation with SARS-CoV-2 peptides. Combined together, Figure 3 gives a holistic representation of the CD4+ T cell responses in the two groups of patients, which we think is important. Therefore, we decided to keep it as a main figure to reflect its importance.

Monocytes and neutrophils: the presented data are very convincing about the differences within monocyte subclasses and their sensitivity to type I IFN. Fig 4D: it would be informative to describe in the legend, how the gene expression in different cell subpopulations were compared between asymptomatic vs symptomatic patients: subtraction, ratio, please clarify.

Response: The gene expression in inflammatory monocytes and neutrophils were compared between asymptomatic vs symptomatic patients by ratio with respect to asymptomatic patients. DEGs highlighted in Figure 5D (Previously Figure 4D) were more enriched in symptomatic patients. To explain the comparison, additional description has been added to the legend in Figure 5D to read "Expression data of genes associated with inflammatory monocytes, activated neutrophils and pro-inflammatory cytokines were compared between acute asymptomatic (n=30) and asymptomatic (n=26) COVID-19 patients and were represented by ratio with respect to expression in asymptomatic patients."

Fig 4E: why the systemic mediators in patients were not compared to healthy controls? Looking at the IL-6 values, it is remarkable to observe an increase in

patients, which are nevertheless largely lower than the levels reported in other causes of acute respiratory failure with ARDS, which can also be discussed in the Discussion section.

Response: The systemic mediator levels of healthy controls were included as black dotted lines within the plots in Figure 5E (Previously Figure 4E).

We are aware of the other meta-analysis studies indicating a lower IL-6 concentration in COVID-19 patients compared to patients with other causes of ARDS (Leisman et al, 2020; Sinha et al, 2020). We thank the reviewer for the suggestion, and have now added this observation into the discussion, on page 15, lines 311 to 314 read “Notably, it has been reported in meta-analysis studies that IL-6 concentrations in COVID-19 patients are significantly lower than patients with ARDS unrelated to COVID-19. This downplays the pro-inflammatory role of IL-6 in severe COVID-19 disease(Leisman et al, 2020; Sinha et al, 2020)”

Citation:

1. Leisman, D. E., Ronner, L., Pinotti, R., Taylor, M. D., Sinha, P., Calfee, C. S., ... & Deutschman, C. S. (2020). Cytokine elevation in severe and critical COVID-19: a rapid systematic review, meta-analysis, and comparison with other inflammatory syndromes. *The Lancet Respiratory Medicine*.
2. Sinha, P., Matthay, M. A., & Calfee, C. S. (2020). Is a “cytokine storm” relevant to COVID-19?. *JAMA internal medicine*, 180(9), 1152-1154

Discussion:

The discussion is well conducted and informative. Some aspect might be seen from different point of view. P14 last {section sign} discussed the calprotectin level lowered in asymptomatic vs the symptomatic patients. If the hypothesis is correct, it can also be seen as DAMPs molecule related to the cell damage as observed in severe septic patients, maintaining a high plasma level. Were differences between the severity (mild, moderate, severe) patients existed?

Response: We thank the reviewer for raising this question. Calprotectin levels are positively correlated with neutrophil counts and severity, as reported by Silvin et. al., 2020. In our cohort, we also found that the transcriptomic levels of S100A6, S100A8 and S100A12 which are subunits of calprotectin, were significantly different between symptomatic and asymptomatic patients (Figure 5D). However, transcriptomic levels of these subunits were not significantly different amongst the symptomatic patients when they were stratified by severity. Results are shown below for review purposes.

Citation:

1. Silvin, A., Chapuis, N., Dunsmore, G., Goubet, A. G., Dubuisson, A., Derosa, L., ... & Solary, E. (2020). Elevated calprotectin and abnormal myeloid cell subsets discriminate severe from mild COVID-19. *Cell*, 182(6), 1401-1418.

P15 {section sign} 2 appears mainly hypothetical and questions the interest to look only at the blood to study the immune response in asymptomatic patients. Even ethically difficult to justify, information from BALavage would be more than important to support the hypothesis. L311 to 318 appears also elusive and could be rewritten.

Response: We note the reviewer's concern and we have re-written this portion to make it clearer. We have also included observational evidence from our transcriptomic findings. We have now removed portions that remain hypothetical with limited value to the discussion. This portion of the discussion, now on page 16, lines 342 to 348, reads "This is consistent with the upregulation in T cell cytotoxic activity related genes observed in asymptomatic patients. Concordantly, *SELL* expression, which is known to be down-regulated upon TCR engagement (Szabo et al, 2019; Ivetic et al, 2019), is also found to be lower in asymptomatic patients. A more efficient T cell antiviral immunity was postulated in a recent study, which showed a highly functional virus-specific CD4+ T cell response in asymptomatic infections (Le Bert et al, 2021)".

P 16 L3275 mentioned a "cytokine storm" that is not demonstrated in severe COVID patients as observed in septic or ARDS patients as recently shown. Consequently, it could be then wrong to consider the cytokine storm as a target for immune-modulation drug, as tocilizumab.

Response: We acknowledge that several meta-analysis studies have downplayed the role of cytokine storms in COVID-19 patients. We have revised the phrasing of this sentence in the discussion on page 17, lines 358 to 361, to read "This would help assess the feasibility of S1P pathway modulation to help limit the respiratory distress and inflammation in COVID-19 patients, since S1P1R agonist has proven successful to treat mice from ARDS during fatal H1N1 infections(Zhao et al, 2019)."

Referee #2 (Remarks for Author):

In the manuscript entitled "Asymptomatic COVID-19: disease tolerance with efficient anti-viral immunity against SARS-CoV-2" by Chan and collaborators, the authors aimed to investigate the immune response profiles underlying asymptomatic SARS-CoV-2 infection. The study was performed on patients recruited for a COVID-19 PROTECT cohort in Singapore. Subjects were classified as symptomatic (n=215) and asymptomatic (n=48), based on a positive SARS-CoV-2 PCR test with or without reported COVID-19 symptoms, respectively. Whole blood was collected and PBMC/plasma used for transcriptomic profiling using the HiSeq 4000 System Illumina technology, multiplex microbead-based immunoassay, CyTOF and Flow cytometry analysis, and antibody-mediated pseudovirus neutralization assay. The results support the idea that asymptomatic patients display a protective moderate pro-inflammatory adaptive cellular immune response. The authors concluded that asymptomatic patients exhibit disease tolerance.

Results in Figure 1 depict transcriptomic signatures of symptomatic and asymptomatic patients revealed by high throughput RNA sequencing. Differentially expressed genes (DEG) were identified. Principal component analysis (PCA) show a clear segregation between symptomatic and asymptomatic patients. Gene ontology (GO) analysis revealed the biological functions of DEGs over and under expressed in asymptomatic patients. Results in Figure 2 depict the expression of plasma markers quantified by Luminex immunoassay, with a focus on differentially expressed growth factors and pro-inflammatory cytokines. Asymptomatic individuals showed higher levels of certain growth factors (e.g., BDNF) and lower levels of pro-inflammatory mediators. Results in Figure 3 depict T cell-associated DEG, as well as CyTOF and flow cytometry validations of T cell subsets (Naives, TEMRA, CM and EM) and Th1/Th17 profiles. Of note, the % of SARS-CoV-2-specific T cells producing IL-17A was higher in asymptomatic compared to symptomatic patients. In Figure 4 mature/immature neutrophils were quantified by flow cytometry, monocyte subsets were analysed by CyTOF, and Luminex immunoassay was used to quantify cytokines/chemokines. In the Figure 5 results show that symptomatic compared to asymptomatic patients display decreased expression of markers associated with cellular repair and leukocyte migration, such as CCR6, CCR7 and CXCR5 on DC, T cells and B cells. Finally, Figure 6 summarizes the main findings of the study. Briefly, the authors propose a model in which symptomatic infection is a result of immune dysregulation, while asymptomatic patients mount a strong Th17 cell response, a sustained neutralizing antibody activity and effective tissue healing process during acute infection.

The research topic is highly relevant for the current SARS-CoV-2 pandemic. The paper is straightforward and overall easy to read. The study idea of assessing early acute immune responses response is very interesting. Diverse techniques were employed that showed coherent results. However, multiple aspects need to be addressed prior publication, as listed below:

Response: That is an accurate and succinct summary, and we will address the concerns below.

Major criticisms:

1. The results on Ab-mediated neutralisation (Figure EV2) should be presented as a main figure. The authors should also include data on the quantification of these Abs in the plasma (IgG and IgM).

Response: We thank the reviewer for the suggestion. Results from S-flow, which is a flow cytometry-based assay for the sensitive detection of total anti-spike IgG and IgM antibodies (Goh et al, 2021) has been included in Figure 4 (Previously Figure EV2). These results have also been shifted to main Figure 4. Please see the new Figure 4 as attached below. The results has also been revised on page 10, lines 190 to 197, to read “Despite upregulation of Th2 signatures in asymptomatic patients, which have been shown to mediate humoral response against viral infections (Spellberg & Edwards, 2001), the SARS-CoV-2 spike-specific IgG responses were comparable between the symptomatic and asymptomatic patients (Fig 4A), while spike-specific IgM responses were significantly lower in the asymptomatic patients (Fig 4B). Importantly, the anti-SARS-CoV-2 neutralizing capacity was also markedly lower in asymptomatic patients compared to symptomatic patients against SARS-CoV-2 (Fig 4C and 4D)”.

Figure 4. SARS-CoV-2 spike-specific antibody responses and neutralizing capacity of symptomatic and asymptomatic COVID-19 patients. (A) IgG and (B) IgM responses were analysed by screening plasma samples of asymptomatic (n=39, median study day 29) and symptomatic (n=57, median study day 31) COVID-19 patients. Plasma samples of asymptomatic (median study day 29) and symptomatic (median study day 31) COVID-19 were assessed for their anti-SARS-CoV-2 neutralization capacity using luciferase expressing lentiviruses pseudotyped with SARS-CoV-2 spike (S) protein of either the original strain, D614, or the mutant variant, G614. Log₁₀ neutralisation IC₅₀ profiles against (C) D614 and (D) G614 pseudoviruses at 1 month post admission (Asymptomatic, n=49; symptomatic, n=57 (D614) or 55 (G614)). Data represent the mean of two independent experiments and statistical analysis was performed with Welch's *t* test (***)*P*<0.001).

Figure 4, Chan et al., 2021

Citation:

1. Goh, Y. S., Chavatte, J. M., Jieling, A. L., Lee, B., Hor, P. X., Amrun, S. N., ... & Renia, L. (2021). Sensitive detection of total anti-Spike antibodies and isotype switching in asymptomatic and symptomatic individuals with COVID-19. *Cell Reports Medicine*, 2(2), 100193.

2. The authors mention differences in CT values between symptomatic and asymptomatic patients (lines 120-123). However, in Table EV1 these differences were not so dramatic (30 vs 32). What are numbers between brackets? Please indicate the limit of detection and the CT value of the negative control. Also, was plasma viremia measured in the two groups?

Response: RT-PCR was performed on respiratory samples collected from symptomatic and asymptomatic patients as previously described by Young et al, 2020. The numbers in the brackets indicate the CT values. No threshold for the CT value above which a swab result is classified as negative is applied. Plasma viremia was not measured in the two groups as incidence of viremia in COVID-19 is low.

Citation:

1. Young, B. E., Ong, S. W. X., Kalimuddin, S., Low, J. G., Tan, S. Y., Loh, J., ... & Lye, D. C. (2020). Epidemiologic features and clinical course of patients infected with SARS-CoV-2 in Singapore. *Jama*, 323(15), 1488-1494.

3. It will be important to provide a brief definition of the term tolerance, as to avoid inaccurate interpretations. One could interpret tolerance as absence of immune response.

Response: We thank the reviewer for highlighting this possible cause of confusion. We have added the brief definition of disease tolerance in the introduction on page 5, lines 94 to 96, to read "These cases are also scientifically intriguing as they demonstrate disease tolerance, illustrated with an immune response that efficiently resolves the viral infection without apparent pathogenic effects."

4. For all figures showing gene expression the contrast should be clarified and should be the identical throughout the manuscript (e.g., asymptomatic versus symptomatic). Figure legends should clarify color codes used.

Response: We have ensured that the comparison of gene expression are clear and are done in an identical manner throughout the manuscript. Figure legends also contain information on the color codes used.

5. Figure 1: the authors should provide a study design and better explain the inclusion criteria of asymptomatic and symptomatic patients at different times post-admission and post-illness onset. Such differences may account for differences in the results.

Response: A similar concern on the timing differences of blood sample collection from the patients has been addressed for Reviewer 1. We have also added a sentence in the methods on page 19, lines 406 to 408 to explain the inclusion criteria, which reads "Blood was collected in VACUETTE EDTA tubes (Greiner Bio, #455036) for healthy donors and acute patients on the day of admission into hospital, or in Cell Preparation Tubes (CPT) (BD, #362761) for recovered patients at various timepoints."

6. Figure 1C: the complement activation genes are under expressed. This is an important finding not discussed in the manuscript. The authors should discuss how early complement activation monitoring can be used to identify patients at risks of developing severe disease in order to treat them early. The same observation for responses to type I IFN. Could the authors measure plasma or cell associated IFN? Can complement and IFN be better predictors for disease aggravation?

Response: This is an interesting comment and we have amended the Discussion to reflect this on page 15 lines 319 to 325, to read “The transcriptome of asymptomatic patients showed a lower level of systemic complement activation compared to symptomatic patients. Systemic complement activation has been associated with respiratory failure in COVID-19 patients (Holter et al, 2020), and excessive activation of complement contributes to destructive inflammation harming the host (Ricklin & Lambris, 2013). This may explain the milder inflammation in asymptomatic patients and thereby suggests therapeutic modulation of complement activity at the early phase of disease as an attractive intervention for COVID-19.”

We have measured and compared the plasma IFN- α levels between the symptomatic and asymptomatic patients and found no significant difference between the two groups. Notably, we have published a study identifying the cytokines associated with COVID-19 disease severity and aggravation (Young et al, 2020), but type I IFN did not perform well as a prognostic marker.

Citation:

1. Young, B. E., Ong, S. W., Ng, L. F., Anderson, D. E., Chia, W. N., Chia, P. Y., ... & Singapore 2019 Novel Coronavirus Outbreak Research Team. (2020). Viral Dynamics and Immune Correlates of Coronavirus Disease 2019 (COVID-19) Severity. *Clinical Infectious Diseases*.

7. Figure 2A: Is it possible to add healthy control data set in the heatmap?

Response: Yes, the figure has been revised to include healthy controls (n=23). Please see the new revised Figure 2A below.

Figure 2 – Signatures of immune mediators in asymptomatic and symptomatic COVID-19 patients.
A. Heatmap of immune mediator levels in plasma samples of healthy controls (HC) (n=23) and COVID-19 patients who are asymptomatic (n=48), or symptomatic (n=172). First plasma sample from each SARS-CoV-2 PCR-positive patient was extracted for analyses. Each color represents the relative concentration of a particular analyte (blue=low concentration; red=high concentration). Each row represents one patient. Hierarchical clustering was performed on patients in each severity stratum using mEV software.
B. Network analysis of significant immune mediators between symptomatic and asymptomatic COVID-19 patients. Interactive relationships between the cytokines or chemokines were determined by STRING (Search Tool for the Retrieval of Interacting Genes/ Proteins) analysis, with a confidence threshold of 0.5.

Figure 2, Chan et al., 2021

8. Figure EV1 legend, the authors should clarify/rewrite this sentence "First plasma sample from each SARS-CoV-2 PCR-positive patient was extracted for analyses"

Response: We wish to explain that we measured the cytokine and chemokine levels in the plasma samples collected upon patients' recruitment. The sentence in Figure EV1 legend has now been edited to read "Immune mediators were measured in the first plasma sample collected from each SARS-CoV-2 PCR-positive patient."

9. Figure 2A, it is unclear why IL-7 is considered pro-inflammatory?

Response: IL-7 is known for its critical role in development and homeostatic expansion of T and B cells in humans. Other than this, IL-7 has also been reported to induce Th1 and Th17-associated cytokine secretion (Bikker et al, 2012). Moreover, IL-7 has also been shown to induce TNF- α (Van Roon et al, 2008). Multiple studies have demonstrated this finding where IL-7 has a role as an important pro-inflammatory mediator in several chronic (rheumatic) inflammatory autoimmune diseases (Van Roon et al, 2008; Churchman and Ponchel, 2008). Hence, we categorized IL-7 under pro-inflammatory in Figure 2A. More details listed below.

1. Bikker, A., Moret, F. M., Kruize, A. A., Bijlsma, J. W., Lafeber, F. P., & van Roon, J. A. (2012). IL-7 drives Th1 and Th17 cytokine production in patients with primary SS despite an increase in CD4 T cells lacking the IL-7R α . *Rheumatology*, 51(6), 996-1005.
2. Van Roon, J. A. G., Glaudemans, K. A. F. M., Bijlsma, J. W. J., & Lafeber, F. P. J. G. (2003). Interleukin 7 stimulates tumour necrosis factor α and Th1 cytokine production in joints of patients with rheumatoid arthritis. *Annals of the rheumatic diseases*, 62(2), 113-119
3. Churchman, S. M., & Ponchel, F. (2008). Interleukin-7 in rheumatoid arthritis. *Rheumatology*, 47(6), 753-759.

10. Lines179-180: it is unclear how DEGs were classified in Th1, Th2, and Th17

Response: We will combine the responses for pointers 10 and 11 to explain the stratification of Th subsets in Fig 3A. We wish to explain that the classification of the DEGs between symptomatic and asymptomatic patients into Th1-, Th2- and Th17-associated signatures were based off previously reported T cell signatures in the whole blood transcriptomes of COVID-19 patients (Aschenbrenner et al, 2021). GO enrichment and Ingenuity Pathway Analysis were also performed to functionally categorize these DEGs.

To improve clarity, the following text has been added on page 9, lines 184 to 188 to read "These DEGs were categorized and presented as specific T effector cell associated signatures based on previously reported T cell signatures in blood transcriptomes of COVID-19 patients (Aschenbrenner et al, 2021) (Fig 3A). GO enrichment and Ingenuity Pathway Analysis were also performed to functionally categorized these DEGs (Fig 3A)."

Citation:

1. Aschenbrenner, A. C., Mouktaroudi, M., Kraemer, B., Oestreich, M., Antonakos, N., Nuesch-Germano, M., ... & Ulas, T. (2021). Disease severity-specific neutrophil signatures in blood transcriptomes stratify COVID-19 patients. *Genome Medicine*, 13(1), 1-25.

11. Figure 3A: please clarify how the stratification in Th subsets was performed. Curiously, Th2 cells appear to express more Lck and ZAP-70, a characteristic of

Th17 cells. Also, SELL or L-Selectin is lost with cell activation; or here it is expressed at the highest levels in symptomatic patients. Please discuss these findings.

Response: We wish to explain that cytoplasmic tyrosine kinases *LCK* and *ZAP70* have been activated upon TCR binding to cognate ligand (Thill et al, 2016; Nika et al, 2010). Hence, an increased expression of *LCK* and *ZAP70* may reflect an upregulation of TCR signalling in asymptomatic patients. Enhanced TCR signalling may result in a multitude of CTL effector functions (Esser et al, 1996). This is consistent with the upregulation in T cell cytotoxic activity related genes observed in asymptomatic patients.

As for the case of higher SELL expression in symptomatic patients, that is in line with our data on a higher T-cell activation in asymptomatic patients. Indeed, this gene is known to be down-regulated upon TCR engagement (Szabo et al, 2019; Ivetic et al., 2019).

We have further discussed these findings in the discussion section on page 16, lines 336 to 346. It now reads “This is consistent with the higher expression of *LCK* and *ZAP70* in asymptomatic patients, which are associated with TCR signaling and T cell activation (Thill et al, 2016; Nika et al, 2010). Enhanced TCR signaling may also result in a multitude of cytotoxic T lymphocyte functions (Esser et al, 1996). Notably, the immune-regulatory functions of IL-17 produced by Th17 cells, amongst others, are also associated with anti-viral Th1 cell immunity and cytotoxic T cell activity, which may enhance virus clearance and hence limit virus induced damages to the infected host (Schmidt & Varga, 2018; Bagri et al, 2017). This is consistent with the upregulation in T cell cytotoxic activity related genes observed in asymptomatic patients. Concordantly, SELL expression, which is known to be down-regulated upon TCR engagement (Szabo et al, 2019; Ivetic et al, 2019), is also found to be lower in asymptomatic patients.”

Citation:

1. Esser, M. T., Krishnamurthy, B., & Braciale, V. L. (1996). Distinct T cell receptor signaling requirements for perforin-or FasL-mediated cytotoxicity. *The Journal of experimental medicine*, 183(4), 1697-1706
2. Thill, P. A., Weiss, A., & Chakraborty, A. K. (2016). Phosphorylation of a tyrosine residue on Zap70 by Lck and its subsequent binding via an SH2 domain may be a key gatekeeper of T cell receptor signaling in vivo. *Molecular and cellular biology*, 36(18), 2396-2402
3. Nika, K., Soldani, C., Salek, M., Paster, W., Gray, A., Etzensperger, R., ... & Acuto, O. (2010). Constitutively active Lck kinase in T cells drives antigen receptor signal transduction. *Immunity*, 32(6), 766-777
4. Szabo, P. A., Levitin, H. M., Miron, M., Snyder, M. E., Senda, T., Yuan, J., ... & Sims, P. A. (2019). Single-cell transcriptomics of human T cells reveals tissue and activation signatures in health and disease. *Nature communications*, 10(1), 1-16.
5. Ivetic, A., Hoskins Green, H. L., & Hart, S. J. (2019). L-selectin: a major regulator of leukocyte adhesion, migration and signaling. *Frontiers in immunology*, 10, 1068

12. Figure 3: It is surprising that in asymptomatic patients, effector memory CD4 T

cells decreased compared to symptomatic but in DEG analysis there was an upregulation of T cell proliferation. Authors need to justify this result. Also, authors should graphically represent the data for both study groups for Ki67 and activation marker for both CD4 and CD8 subsets. It will be important to understand how cell proliferation and activation state is changing with disease severity.

Response: Unfortunately, we do not have Ki67 markers in our panels to correlate the transcriptomic data to the cellular data. However, we wish to explain that the transcriptomic analyses highlighted increase in proliferation and TCR signaling DEGs in asymptomatic patients (Figure 3A). TCR signaling is essential for activation, proliferation, and effector function of T cells. Early induction of functional T cells associates with rapid viral clearance and mild disease in COVID-19 patients (Tan et al., 2020). This indicates early induction of functional T cells in asymptomatic cases, which may enhance virus clearance and hence limit virus induced damages to the infected host. Although the number of effector memory CD4 T cells are lower in asymptomatic patients, there is a possibility that a larger proportion of these T cells are specific against the SARS-CoV-2 and at a more activated state. This hypothesis is supported by another study, whereby they demonstrated that asymptomatic patients mount a highly functional virus-specific T cell response against SARS-CoV-2 (Le Bert et al, 2021).

Citation:

1. Tan, A. T., Linster, M., Tan, C. W., Le Bert, N., Chia, W. N., Kunasegaran, K., ... & Bertoletti, A. (2021). Early induction of functional SARS-CoV-2-specific T cells associates with rapid viral clearance and mild disease in COVID-19 patients. *Cell reports*, 34(6), 108728.
2. Le Bert, N., Clapham, H. E., Tan, A. T., Chia, W. N., Tham, C. Y., Lim, J. M., ... & Tam, C. C. (2021). Highly functional virus-specific cellular immune response in asymptomatic SARS-CoV-2 infection. *Journal of Experimental Medicine*, 218(5).

13. Figure 3C: Although the idea that asymptomatic compared to symptomatic patients develop more robust SARS-CoV-2-specific Th17 responses is interesting, the number of participants in these comparisons is too low to allow conclusions. The authors should provide sample size calculations for comparisons between groups. Does plasma IL-17A levels also vary in the same direction?

Response: We wish to explain that the SARS-CoV-2 specific responses in T cells were performed on a subset of the patients in this cohort to further characterize the differences in T effector response between symptomatic and asymptomatic patients. As it was a retrospective study, we selected the patients based on availability of first convalescent sample at around the similar study day. This is a limitation that we have included in the methods section, page 24, lines 512 to 514 reflect this. This reads "As the comparison of SARS-CoV-2 specific T cell responses between symptomatic and asymptomatic patients was retrospective in nature, samples were selected for comparison based on matching study day, and sample availability of the PBMCs." There were no significant differences in plasma IL-17A levels between symptomatic and asymptomatic patients.

14. Are there other clinical parameters available, such as C Reactive Protein and d-Dimer levels, to allow a better correlation between study findings and pre-existing conditions?

Response: Yes, other clinical parameters of the COVID-19 patients at the time of admission are available, including C-reactive protein and LDH levels, which has been associated with severe COVID-19. Unfortunately, d-dimer levels are not available. We have included the clinical parameters into Table EV1. The comparison of CRP and LDH levels between symptomatic and asymptomatic patients is also mentioned in the discussion on page 7, lines 123 to 125, to read “Notably, symptomatic patients had significantly higher C reactive protein and lactate dehydrogenase levels compared to asymptomatic patients, indicating higher levels of inflammation (Poggiali et al, 2020).”

15. Discussion: One interesting finding is BDNF. Can a decrease in BDNF explain the neuropathogenesis of SARS-CoV-2 in symptomatic patients?

Response: The association of BDNF and neuropathogenesis is not clear in our cohort, as we did not observe any patients with neuropathology during the acute and early convalescent phase of COVID-19. However, we agree that this is an interesting hypothesis, as increasing BDNF levels have been associated with COVID-19 recovery (Azoulay et al, 2020).

1. Azoulay, D., Shehadeh, M., Chepa, S., Shaoul, E., Barhom, M., Horowitz, N. A., & Kaykov, E. (2020). Recovery from SARS-CoV-2 infection is associated with serum BDNF restoration. *The Journal of infection*.

16. Discussion: the application of this study is not only to characterize asymptomatic patients but to identify biomarkers that can as early as possible predict severe disease evolution in symptomatic patients. Such predictors should be discussed (complement, IFN).

Response: We thank the reviewer for the suggestion. We do not think that the biomarkers highlighted in this cohort study will be better and more efficient prognosis factors than the ones reported in two of our previous studies (Carissimo et al, 2020; Young et al, 2020). These previous studies have already identified several early prognostic biomarkers like immature neutrophil counts and a few cytokines that could predict severe disease evolution in symptomatic patients, notably from a subset of the symptomatic patients included in this study. Hence, we did not include the element of prognosis into this study, but instead chose to focus on the cross-sectional comparison between symptomatic and asymptomatic patients at the early acute and early convalescent time points.

Citations:

1. Carissimo, G., Xu, W., Kwok, I., Abdad, M. Y., Chan, Y. H., Fong, S. W., ... & Ng, L. F. (2020). Whole blood immunophenotyping uncovers immature neutrophil-to-VD2 T-cell ratio as an early marker for severe COVID-19. *Nature communications*, 11(1), 1-12.
2. Young, B. E., Ong, S. W., Ng, L. F., Anderson, D. E., Chia, W. N., Chia, P. Y., ... & Singapore 2019 Novel Coronavirus Outbreak Research Team. (2020). Viral Dynamics and Immune Correlates of Coronavirus Disease 2019 (COVID-19) Severity. *Clinical Infectious Diseases*.

17. Discussion: Authors could further discuss the impact of the study's result, on how it could be used to find therapeutic targets to fight COVID-19.

Response: We thank the reviewer for the suggestion. We have added into the discussion section on page 17 lines 368 to 371, to read “Developing or repurposing therapy that can rectify the immune dysregulation, including limiting inflammation (TRC Group, 2020; Kalil et al, 2020; Ong et al, 2020) or boosting T cell responses (Le Bert et al, 2021), are viable options to limit COVID-19 progression.” to emphasize the impact of this study.

Minor criticisms:

1. Although in lines 115-120 the authors describe the ethnicity of the participants, in Table E1, authors need to include other Ethnicities, not only Chinese.

Response: Other large ethnicities in Singapore have been included into Table EV1. Ethnicities of the participants in this study are now subdivided into Chinese, Malay, Indian, and others.

2. Please write T_HX and not ThX

Response: All subscripts for helper T cell acronym “Th” have been removed.

3. On the line 69, to clarify what is together: the data or the patients?

Response: It refers to the data. The sentence has been corrected to read “Together, the data suggest that asymptomatic patients mount less pro-inflammatory and more protective immune responses against SARS-CoV-2 indicative of disease tolerance.”

4. On the line 272, the sentence should be reorganized since it the word transcriptomic looks like it's detached.

Response: The sentence has been amended to read “In order to understand why SARS-CoV-2 infection are asymptomatic in a fraction of patients, we analyzed in detail the transcriptomic signatures, neutralizing capacity of antibodies, cellular phenotypes and soluble immune profiles of COVID-19 patients from a Singapore cohort”

5. On the line 296: "a another" English mistake.

Response: Typographical error has been corrected to read “Our data are in agreement with another study using a cohort of patients from China (Long et al, 2020).”

6. On the line 332: "pathologie" English mistake.

Response: Typographical error has been corrected to “pathologies”.

Referee #3 (Comments on Novelty/Model System for Author):

There have already been a few papers on the immune profiles of asymptomatic COVID-19 patients, though this study is likely the most comprehensive. The study is confounded by differences in the two groups (ethnicity, timing of sample collection, etc) that limit some of the findings.

Referee #3 (Remarks for Author):

Chan et al. present a detailed immunologic study of asymptomatic COVID-19 subjects compared to symptomatic subjects. They obtained peripheral blood at convalescent timepoints and use a variety of complementary techniques to characterize the overall host response (analysis of blood transcriptomes, Luminex of serum cytokines and growth factors, mass and flow cytometry) as well as the anti-viral response (neutralization assays, intracellular cytokine staining after peptide pool stimulation). The main findings are that "asymptomatic patients mount less pro-inflammatory and more protective immune responses against SARS-CoV-2 indicative of disease tolerance." Overall, I found the presentation of data in main and supplemental figures to be complete and thorough and the paper easy to read. The methods are thoroughly described and statistical analyses are mostly justified. To my knowledge, the paper represents one of the more thorough immunologic analyses of asymptomatic COVID-19 patients. My suggestions are mostly focused on interpretation of the findings and some minor clarifications.

A significant limitation in the study design is that a minority of the asymptomatic subjects were ethnic Chinese and likely had different exposures by virtue of their lifestyles and profession. Thus, the results may be confounded by both genetics and exposure history. This should be discussed as a limitation, especially in the context of emerging data that genetics plays a role in COVID-19 outcomes (including GWAS studies).

Response: We thank the reviewer for raising the limitation of ethnic representation. Information on the ethnicity has been expanded in Table EV1 to illustrate the major ethnic groups of the Singapore population to include Chinese, Malays, Indians and others in the symptomatic and asymptomatic patients. We have also included the limitation on page 14, lines 288 to 291, in the discussion, to read "Notably, our cohort contains ethnic differences between asymptomatic and symptomatic patients (Table EV1), which could possibly influence the findings due to existing variance in genetic factors (Sze et al, 2020). However, these differences could also be limited as our cohort are of majority Asian ethnicity."

Another source of confounding is the timing of sample collection, which was uniformly earlier among asymptomatic subjects. This should also be discussed as a limitation.

Response: A similar concern on the timing of blood sample collection from the patients has been addressed for Reviewers 1 and 2 above.

The Discussion seeks to claim that the data explain "why SARS-CoV-2 infections are asymptomatic" (Line 271). But the study is cross-sectional, not longitudinal, and uses samples collected after symptom onset, not before. Thus, any language about

causal inference is an over interpretation of the data presented. Rather, the authors have only shown associations (albeit important ones), and this should be clearly acknowledged throughout the text. On that note, Figure 6 is purely speculative and should be removed entirely.

Response: We agree that our study is a cross-sectional comparison between symptomatic and asymptomatic patients, and have no intention to definitively identify factors that lead to pathology or protection against COVID-19. We have re-emphasized the purpose of our study on page 14, lines 284 to 288, to read “In order to understand why SARS-CoV-2 infection is asymptomatic in a fraction of patients, we performed a cross-sectional comparison of the transcriptomic signatures, neutralizing capacity of antibodies, cellular phenotypes and soluble immune profiles of COVID-19 patients from a Singapore cohort. This could help identify immune responses that are associated with pathology or protection against COVID-19.”. Additional care has been taken to prevent over-interpretation of the data in the discussion. Although our intention for figure 6 is to provide a overview of our findings, it has been removed from the manuscript to prevent over speculation of the underlying mechanisms of COVID-19.

Minor

It is unclear to me why asymptomatic patients were admitted to the hospital.

Response: A similar concern on the admission of asymptomatic patients into dedicated community facilities for quarantine has been addressed for Reviewer 1.

Fig EV1. IL-1 β and IL-18 are not Th1 cytokines. In fact, it is much more likely that they were produced by myeloid cells prior to detection in plasma.

Response: We have corrected Fig EV1 to reflect this. IL-1 β and IL-18 have been re-categorized from T cell associated immune mediators to inflammatory macrophage associated cytokines in Fig EV1C.

Fig EV3. It is inaccurate to define CD4+CD25+ T cells as Tregs. CD25 is expressed on activated T cells, and it is well known that this definition does not fully correlate with FoxP3 staining.

Response: We agree that CD25 expression does not fully correlate with FoxP3 staining, and hence cannot act as a sole marker for the definition of Tregs. Hence, we have renamed this population to be CD127- CD25+ cells on Appendix Fig S1 (Previously Fig EV3).

Fig EV3. Gating for Granzyme B is not shown.

Response: Thank you for raising up the missing gating strategy. We have included a representative gating strategy to isolate the Granzyme B+ CD8+ T cell population in Fig EV2B, as attached below.

Figure EV2 – CD8+ T cells responses in COVID-19 patients.

A. Mass cytometry was performed on PBMCs obtained from acute symptomatic (n=37) and acute asymptomatic (n=19) COVID-19 patients. Naive, TEMRA, Central memory (CM) and effector memory (EM) T cells were characterized based on CD45RA and CCR7 expressions. Data are presented as Mean \pm SD *P<0.05; ***P<0.001 (Kruskal-Wallis test with Dunn's multiple comparison).

B. Representative gating strategy for the characterization of granzyme B expression of CD8+ T cells in isolated PBMCs of COVID-19 patients by flow cytometry. Representative gating strategy was performed on a symptomatic patient.

C. Comparison of granzyme B expression in CD8+ T cells from symptomatic (n=5) and asymptomatic (n=5) convalescent PBMCs. Data are presented as Mean \pm SD (Mann-Whitney U test).

Figure EV2, Chan et al., 2021

Table EV3 lists 29 markers as part of a flow cytometry panel that was run on an

LSRII cytometer (Line 456). This is technically impossible, I believe. Was the instrument a BD Symphony or Aurora Cytometer?

Response: We thank the reviewer for this comment. Samples stained for the 29 markers were acquired by the Cytex Aurora cytometer. We have included the missing methods into the materials and methods section, on pages 23 and 24, lines 502 to 514. We have also included the missing panel used to profile the T effector subsets in Appendix Table S1 (Previously Table EV3), under Intracellular Panel.

23rd Apr 2021

Dear Prof. Ng,

Thank you for the submission of your revised manuscript to EMBO Molecular Medicine. I am pleased to inform you that we will be able to accept your manuscript pending the following final amendments:

- 1) Figures: Please remove legends from all figure files. Figure legends should be only in the main manuscript file.
- 2) In the main manuscript file, please do the following:
 - Correct/answer the track changes suggested by our data editors by working from the attached/uploaded document.
 - Remove text highlight color.
 - Add callouts for Fig EV1 A, B, C. Also, Fig 3B, C should be called out before Fig 4. Please make sure that figures are called out in a sequential order.
 - Make sure that all special characters display well.
 - In M&M, include that, in addition to the WMA Declaration of Helsinki, the experiments conformed to the principles set out in the Department of Health and Human Services Belmont Report.
 - In addition to the accession number please provide URL for nucleotide and amino acid sequences of RSV neutralizing antibodies. Please be aware that all datasets should be made freely available upon acceptance, without restriction. Use the following format to report the accession number of your data:

[data type]: [full name of the resource] [accession number/identifier] ([doi or URL or identifiers.org/DATABASE:ACCESSION])

Please check "Author Guidelines" for more information.

<https://www.embopress.org/page/journal/17574684/authorguide#availabilityofpublishedmaterial>

- 3) Funding: Please remove "Funding" section and list all sources of funding in "Acknowledgements".
- 4) Synopsis image: Please provide the image as a separate, high-resolution jpeg file 550 px-wide x (250-400)-px high to illustrate your article.
- 5) Press release: Please inform us as soon as possible and latest at the time of submission of the revised manuscript if you plan a press release for your article so that our publisher could coordinate publication accordingly.
- 6) Please be aware that we use a unique publishing workflow for COVID-19 papers: a non-typeset PDF of the accepted manuscript is published as "Just Accepted" on our website. With respect to a possible press release, we have the option to not post the "Just Accepted" version if you prefer to wait with the press release for the typeset version. Please let us know whether you agree to publication of a "Just accepted" version or you prefer to wait for the typeset version.
- 7) As part of the EMBO Publications transparent editorial process initiative (see our Editorial at <http://embomolmed.embopress.org/content/2/9/329>), EMBO Molecular Medicine will publish online a Review Process File (RPF) to accompany accepted manuscripts. This file will be published in conjunction with your paper and will include the anonymous referee reports, your point-by-point response and all pertinent correspondence relating to the manuscript. Let us know whether you agree with the publication of the RPF and as here, if you want to remove or not any figures from it prior to publication. Please note that the Authors checklist will be published at the end of the RPF.

8) Please provide a point-by-point letter INCLUDING my comments as well as the reviewer's reports and your detailed responses (as Word file).

I look forward to reading a new revised version of your manuscript as soon as possible.

Yours sincerely,

Zeljko Durdevic

***** Reviewer's comments *****

Referee #1 (Comments on Novelty/Model System for Author):

"Adequate" was used instead of high, because the pandemic context and work overload precluded organized and time delay for performing experiments. It is then not a limitation for the research itself.

Referee #1 (Remarks for Author):

The Reviewer thanks the Authors for the easy to follow responses provided by the Authors. Almost all raised points have been answered, improved when it was possible, and clarified when it was not.

Referee #2 (Comments on Novelty/Model System for Author):

This is a revised manuscript. The authors addressed all criticism raised by this reviewer.

Referee #2 (Remarks for Author):

The authors addressed all criticism raised by this reviewer.

The authors performed the requested editorial changes.

Corresponding Author Name: Lisa F.P. Ng

Manuscript Number: EMM-2021-14045